# Allosteric modulators enhance agonist efficacy by increasing the residence time of a GPCR in the active state

Anne-Marinette Cao [1,4,5], Robert B. Quast [1,5], Fataneh Fatemi[1,2], Philippe Rondard [3], Jean-Philippe Pin [3✉] & Emmanuel Margeat [1✉]

Much hope in drug development comes from the discovery of positive allosteric modulators (PAM) that display target subtype selectivity and act by increasing agonist potency and efficacy. How such compounds can allosterically influence agonist action remains unclear. Metabotropic glutamate receptors (mGlu) are G protein-coupled receptors that represent promising targets for brain diseases, and for which PAMs acting in the transmembrane domain have been developed. Here, we explore the effect of a PAM on the structural dynamics of mGlu2 in optimized detergent micelles using single molecule FRET at sub-millisecond timescales. We show that glutamate only partially stabilizes the extracellular domains in the active state. Full activation is only observed in the presence of a PAM or the $G_i$ protein. Our results provide important insights on the role of allosteric modulators in mGlu activation, by stabilizing the active state of a receptor that is otherwise rapidly oscillating between active and inactive states.

[1] Centre de Biologie Structurale (CBS), Univ. Montpellier, CNRS, INSERM, Montpellier, France. [2] Protein Research Center, Shahid Beheshti University, Tehran, Iran. [3] Institut de Génomique Fonctionnelle, Univ. Montpellier, CNRS, INSERM, Montpellier, France. [4] Present address: École Polytechnique Fédérale de Lausanne (EPFL), SB ISIC LCBM, Lausanne, Switzerland. [5] These authors contributed equally: Anne-Marinette Cao, Robert B. Quast. ✉email: jean-philippe.pin@igf.cnrs.fr; margeat@cbs.cnrs.fr

G protein-coupled receptors (GPCR) constitute the largest family of integral membrane receptors encoded in the human genome and are involved in various physiological processes[1]. They constitute the main targets in drug development programs for many therapeutic applications[2]. In recent years, much hope came from the discovery of allosteric modulators targeting GPCRs with a few already on the market[3]. Their pharmacological interest comes from their ability to target allosteric sites different from the evolutionary conserved orthosteric site, conferring higher subtypes selectivity. Most importantly, positive allosteric modulators (PAM) enhance agonists effects on GPCRs, then preserving their rhythm of biological activity where and when needed physiologically[4]. PAMs can display various effects[5], including increasing agonist potency (from 2 to 100 fold), increasing agonist efficacy, partially activating receptors (ago-PAM effect), or even orienting the receptor towards one of its signaling pathway[6,7]. It is commonly considered that PAMs act by stabilizing a specific conformation of the receptor[8–12]. However, PAMs may likely act by influencing the equilibrium between preexisting GPCR conformational states.

Class C GPCRs are especially amenable for allosteric modulation, notably given their highly modular architecture, being more complex than the simple rhodopsin-like structure[13]. These receptors include the metabotropic glutamate (mGlu), the GABA (GABA$_B$R), the calcium-sensing (CaSR), and the umami and sweet taste receptors (T1R)[14]. The mGlu receptors are responsible for the modulatory activity to L-glutamate (Glu), the major excitatory neurotransmitter in the central nervous system, and are therefore essential in the fine-tuning of synapses[15]. Class C GPCRs are composed of two subunits, each comprising several functional domains (Fig. 1). The large extracellular domain (ECD) consists of a Venus flytrap domain (VFT), harboring the orthosteric site, and a rigid linker connected to the 7 transmembrane domain (7TM)[8,16,17]. Most identified class C GPCR allosteric modulators act in the 7TM at a site corresponding to the orthosteric site of the rhodopsin-like GPCRs (Fig. 1)[18–21]. Other sites have also been identified close to the orthosteric site[22,23], at the active interface of the VFT dimer[24], or at the active interface of the 7TM dimer[8]. Despite our knowledge of their binding mode, how such molecules allosterically control agonist affinity or efficacy, exert partial agonist activity or biased effect remains largely unknown.

In the present study, we examine the effects of the 7TM-targeting mGlu2 PAM BINA on the conformational dynamics of the receptor at the single-molecule level. Although a few studies reported on the structural dynamics of mGlu receptors on single molecules[25–27], none examined the allosteric modulation by small molecules or G proteins. Here, we optimized the conditions to conserve PAM activity of the solubilized full-length human mGlu2 receptor, N-terminally labeled through a SNAP-tag in each of the subunits, and measured single-molecule Förster resonance energy transfer (smFRET) at nanoseconds time resolution. We show that mGlu2 is oscillating between inactive and active states at submillisecond timescales in its apo state and that Glu partially increased the fraction of receptors residing in the active state. Only in the presence of BINA can the full population of receptors be stabilized in an active conformation, providing a striking explanation for the increased agonist efficacy and potency observed with this PAM. We observe a similar effect with the nucleotide-free G$_i$ heterotrimeric protein. Altogether, the quantification of submillisecond structural dynamics of soluble, functional, full-length mGlu2 receptors sheds light on the mechanism of action of a synthetic mGlu2 PAM and the stabilizing effect of the G$_i$ protein.

## Results

**Optimization of detergent conditions to obtain fully functional mGlu2 dimers.** Our approach to perform smFRET measurements with submillisecond resolution requires fluorescently labeled receptors to be freely diffusing in solution, while maintaining full functional integrity for several hours at room temperature. Therefore, we evaluated a set of different detergents commonly used for GPCR-solubilization, supplemented or not with the cholesterol analogue cholesteryl hemisuccinate (CHS), for their ability to extract receptors from membranes and maintain them in solution, while preserving native-like ligand responsiveness. For this initial detergent screening we employed lanthanide resonance energy transfer (LRET)[28,29], which monitors the VFT N-termini reorientation upon activation (Fig. 1), and was previously reported as an efficient approach to study mGlu structural dynamics[29–32]. We labeled the extracellular N-termini of the mGlu2 subunits using the SNAP-tag technology on HEK293T cells (Fig. 2). This approach does not interfere with receptor function and by using cell-impermeable SNAP-tag substrates only cell surface receptors are labeled, resulting in a homogenous, fully processed, dimeric, fluorescently labeled receptor population[30].

The functional integrity of receptor preparations was assessed upon Glu stimulation in detergent micelles after various time points up to 24 h at room temperature and compared with control conditions of mGlu2 in crude membranes. In parallel, the integrity of the transmembrane domain was evaluated through the effect of BINA (Fig. 2). Indeed, the PAM-binding site is

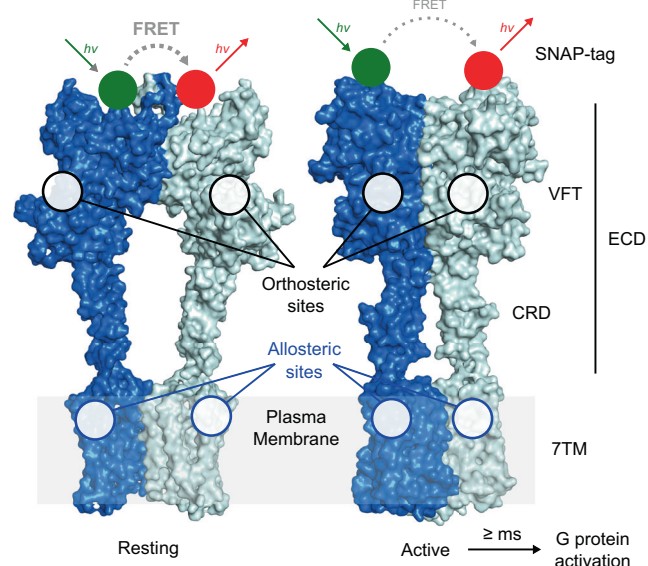

**Fig. 1 Structure and conformational rearrangements of mGlu receptor.** Structures of dimeric mGlu2 in resting and active conformations. The major structural elements of each subunit include the extracellular domain (ECD, comprising the Venus fly-trap domain (VFT) and the cysteine-rich domain (CRD)) and the seven-transmembrane domain (7TM). Orthosteric ligand binding sites are found in the cleft between the upper and lower lobes (black circles) of the VFT and the majority of allosteric modulators bind to sites in the 7TM (blue circles). Activation leads to a closure of the VFTs and a reorientation of the ECDs, the CRDs, and the 7TMs bringing the two subunits into closer proximity. In N-terminally SNAP-tag labeled receptor dimers this leads to a transition from a high FRET/resting to a low FRET/active state. G protein activation through interactions with the cytoplasmic side of the 7TM is reported to occur at >10 ms timescales. The shown structures were generated using PDB ID 7EPA and 7E9G.

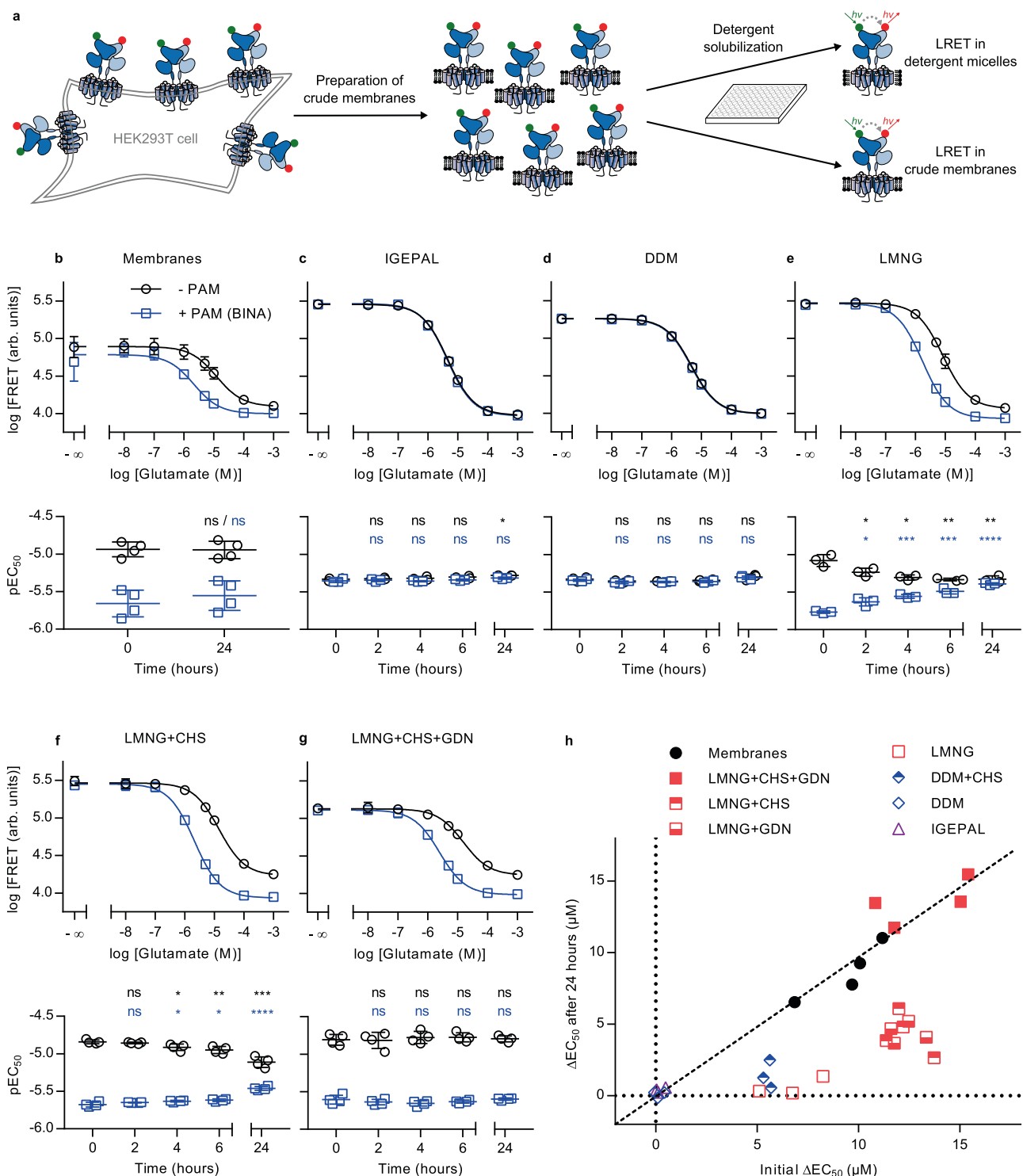

known to be located within the 7TM region[33] and thus the functional link translating the PAM effect to Glu potency at the VFT level provides a reliable measure of the receptor's global functional integrity.

Dose-dependent response to Glu, reflected by a decreasing LRET signal, was observed under all tested conditions but in some cases revealed changes in the Glu $pEC_{50}$ values over time (Fig. 2b–h, S1–12). More importantly, the effect of BINA on Glu potency strongly depended on the detergent mixture used, thus indicating detergent-dependent integrity of the functional link between VFT and 7TM. Positive allosteric modulation was not

observed using IGEPAL (octylphenoxypolyethoxyethanol) and DDM (n-dodecyl β-D-maltoside), two nonionic detergents that have previously been used to solubilize full-length mGlu for smFRET by TIRF microscopy[25] (Fig. 2c–d, h, S1, and S2). Only a weak allosteric modulation by BINA was found when DDM was supplemented with CHS, known to facilitate functional GPCR solubilization through mixed sterol-detergent micelles[34], but this effect was lost within 4–6 h (Fig. 2h, S3). Similarly, allosteric modulation in micelles composed of the branched nonionic detergent LMNG (lauryl maltose neopentyl glycol) was not stable (Fig. 2e, h, S4). In contrast, the addition of CHS to LMNG led to

**Fig. 2 Evaluation of detergents for functional solubilization of full-length mGlu2 using LRET. a** SNAP-mGlu2 dimers were labeled with cell-impermeable lanthanide donor and "green" acceptor fluorophores on living HEK293T cells. After preparation of crude membrane fractions, LRET measurements were performed in microtiter plates either directly on membranes or after detergent solubilization. **b–h** The functional integrity of SNAP-labeled receptors was monitored over time at room temperature based on the dose-dependent intersubunit LRET changes in response to the orthosteric agonist Glu (- PAM) and in combination with 10 µM positive allosteric modulator BINA (+ PAM). **b–g** Dose-response curves at time 0 h (top) and time course of $pEC_{50}$ values (bottom) obtained on crudes membranes (**b**, $n = 4$), in IGEPAL (**c**, $n = 3$), DDM (**d**, $n = 3$), LMNG (**e**, $n = 3$), LMNG-CHS (**f**, $n = 4$) and LMNG-CHS-GDN (**g**, $n = 4$). Data represent the mean from different biological replicates ± SD. Statistical differences of $pEC_{50}$ values for Glu (black) and Glu + BINA (blue) compared to time 0 h were determined using two-sided unpaired $t$-tests and are given as: (**b**) $p_{Glu-24h} = 0.92$ (ns), $p_{Glu+BINA-24h} = 0.46$ (ns), $n = 4$ independent biological samples examined over 3 independent experiments; (**c**) $p_{Glu-2h} = 0.65$ (ns), $p_{Glu-4h} = 0.49$ (ns), $p_{Glu-6h} = 0.14$ (ns), *$p_{Glu-24h} = 0.014$, $p_{Glu+BINA-2h} = 0.68$ (ns), $p_{Glu+BINA-4h} = 0.72$ (ns), $p_{Glu+BINA-6h} = 0.46$ (ns), $p_{Glu+BINA-24h} = 0.086$ (ns), $n = 3$ independent biological samples examined over 3 independent experiments; (**d**) $p_{Glu-2h} = 0.29$ (ns), $p_{Glu-4h} = 0.20$ (ns), $p_{Glu-6h} = 0.46$ (ns), $p_{Glu-24h} = 0.38$ (ns), $p_{Glu+BINA-2h} = 0.0995$ (ns), $p_{Glu+BINA-4h} = 0.068$ (ns), $p_{Glu+BINA-6h} = 0.25$ (ns), $p_{Glu+BINA-24h} = 0.086$ (ns), $n = 3$ independent biological samples examined over 3 independent experiments; (**e**) *$p_{Glu-2h} = 0.048$, *$p_{Glu-4h} = 0.011$, **$p_{Glu-6h} = 0.0057$, **$p_{Glu-24h} = 0.009$, *$p_{Glu+BINA-2h} = 0.014$, ***$p_{Glu+BINA-4h} = 0.00025$, ***$p_{Glu+BINA-6h} = 0.00024$, ****$p_{Glu+BINA-24h} = 0.000008$, $n = 3$ independent biological samples examined over 3 independent experiments; (**f**) $p_{Glu-2h} = 0.33$ (ns), *$p_{Glu-4h} = 0.027$, **$p_{Glu-6h} = 0.0063$, ***$p_{Glu-24h} = 0.00035$, $p_{Glu+BINA-2h} = 0.16$ (ns), *$p_{Glu+BINA-4h} = 0.039$ (*), *$p_{Glu+BINA-6h} = 0.018$, ****$p_{Glu+BINA-24h} = 0.00004$; $n = 4$ independent biological samples examined over 1 independent experiment; **g** $p_{Glu-2h} = 0.89$ (ns), $p_{Glu-4h} = 0.6$ (ns), $p_{Glu-6h} = 0.54$ (ns), $p_{Glu-24h} = 0.78$ (ns), $p_{Glu+BINA-2h} = 0.39$ (ns), $p_{Glu+BINA-4h} = 0.2$ (ns), $p_{Glu+BINA-6h} = 0.43$ (ns), $p_{Glu+BINA-24h} = 0.83$ (ns), $n = 4$ independent biological samples examined over 3 independent experiments. **h** Scatter plot of $\Delta EC_{50}$, i.e. the difference in $EC_{50}$ obtained in presence and absence of BINA, at time 24 h at RT ($y$-axis) vs. at time 0 h ($x$-axis), for membrane fractions and detergent mixtures. The conditions along the diagonal represent those experiencing the lowest changes over time. Source data of panels **b–h** are provided as a source data file.

prolonged functional integrity of the receptors, lasting from 6–24 h, in a CHS-dose-dependent fashion (Fig. 2f, h, S5–7).

The functional integrity of mGlu2 was further improved by the addition of GDN (glyco-diosgenin) to the LMNG-CHS mixture (Fig. 2g, h, S8–10). This steroid-based amphiphile has been demonstrated to improve GPCR stability[35] and was recently employed in structure determination of mGluR5 by cryo-EM[16]. GDN was found beneficial at all concentrations tested (Fig. S8–10), but the presence of CHS remained crucial for the long-term functional integrity of solubilized receptors in micelles (Fig. 2h, S11). Strikingly similar results were obtained for the full-length rat mGlu2 (Fig. S23) previously used in smFRET studies[25,26].

Overall, our results demonstrated that the optimized LMNG-CHS-GDN mixture (0.005% w/v, 0.0004% w/v, and 0.005% w/v, respectively) is mandatory to maintain the functional integrity and allosteric link between the mGlu2 VFT and 7TM domains, for at least 24 h at room temperature (Fig. 2g-h, S9). The LRET signal range as well as the $pEC_{50}$ values for Glu and Glu + PAM in this mixture were well in agreement with those obtained in crude membranes (Fig. 2b and S12), and also reflected earlier observations in live cells[30,31]. Under these detergent conditions, a small but significant effect of the negative allosteric modulator (NAM) Ro64-5229 that reduces the $pEC_{50}$ of glutamate in membranes (Fig. S12a–c) and live cells[30], was observed as well (Fig. S9a–c). In addition, we confirmed the effect of the partial agonist DCG-IV, previously shown to promote changes in the VFT intersubunit orientation to a lower extent than full agonists[30]. Such partial effect, strongly potentiated by the addition of BINA, was indeed observed on the solubilized receptor (Fig. S13). Notably, the effects of DCG-IV in the absence or presence of NAM and PAM reflected those observed in crude membranes (Fig. S12d–f), thus confirming native-like ligand responsiveness of the detergent-solubilized receptor.

For all our data, we noted that the amplitude of the LRET change were lower and the error in $EC_{50}$ larger in membrane fractions as compared to receptors in detergent preparations. We believe that this effect stems from the presence of residual glutamate in the membrane fractions that is eliminated after receptor solubilization in detergent. Such an effect would be sufficient to explain the lower LRET efficiency (as the agonist decreases LRET efficacy), associated with the large variability observed in membranes with no glutamate added (as the amount

of residual glutamate is not controllable and likely variable between membrane preparations).

Note that the GDN and CHS concentrations we used remained sufficiently moderate to not create a fluorescence background that would have been detrimental to our smFRET studies. Indeed, we found that detergent solutions were slightly contaminated with fluorescent species of unknown origin (also found in batches from different suppliers).

## Allosteric modulation through the 7TM is required to stabilize the fully active VFT state.

We then turned to the single-molecule study of full-length mGlu2 and therefore substituted the LRET fluorophores with SNAP-tag substrates of Cy3B as donor and d2 (a Cy5 derivative) as acceptor. Thanks to the pulsed interleaved excitation (PIE)/nanosecond alternating laser excitation (nsA-LEX) confocal configuration, which we previously employed to study isolated mGlu VFTs[31,32], single molecules are detected as they diffuse through the confocal observation volume (Fig. 3a). A 2D histogram representation was used, where the X-axis represents the apparent FRET efficiency ($E_{PR}$) and the Y-axis the stoichiometry factor $S$ calculated for each single molecule (Fig. S14)[36,37]. For further analysis, only donor-acceptor (D-A) containing complexes were selected, based on $S$ ($0.3–0.35 < S < 0.6–0.65$). For each single molecule, we further determined its apparent FRET efficiency ($E_{PR}$), the average fluorescence lifetime of the donor in presence of the acceptor ($\tau_{DA}$), and the average excited-state lifetime of the acceptor ($\tau_A$).

FRET histograms of SNAP-labeled, full-length mGlu2 in LMNG-CHS-GDN micelles showed a wide, multimodal distribution (Fig. 3b–h), indicating the co-existence of four main VFT states. These distributions were observed reproducibly by repeating experiments on independent biological replicates of solubilized receptors originating from membrane fractions prepared at different cell passages (Fig. S15). In the absence of ligands, the main population was centered around $E_{PR} \sim 0.6$ (high FRET, HF, yellow), and less well-defined minor populations were present at lower and higher FRET values (Fig. 3b). Upon application of saturating concentrations of Glu, a second major population at low FRET (LF, $E_{PR} \sim 0.34$, green) appeared (Fig. 3c). Such a decrease in FRET was observed in our smFRET study on freely diffusing isolated VFTs[30,31], as well as on immobilized full-length receptors[25]. Nevertheless, in contrast to our observation on isolated VFTs, which showed a complete shift

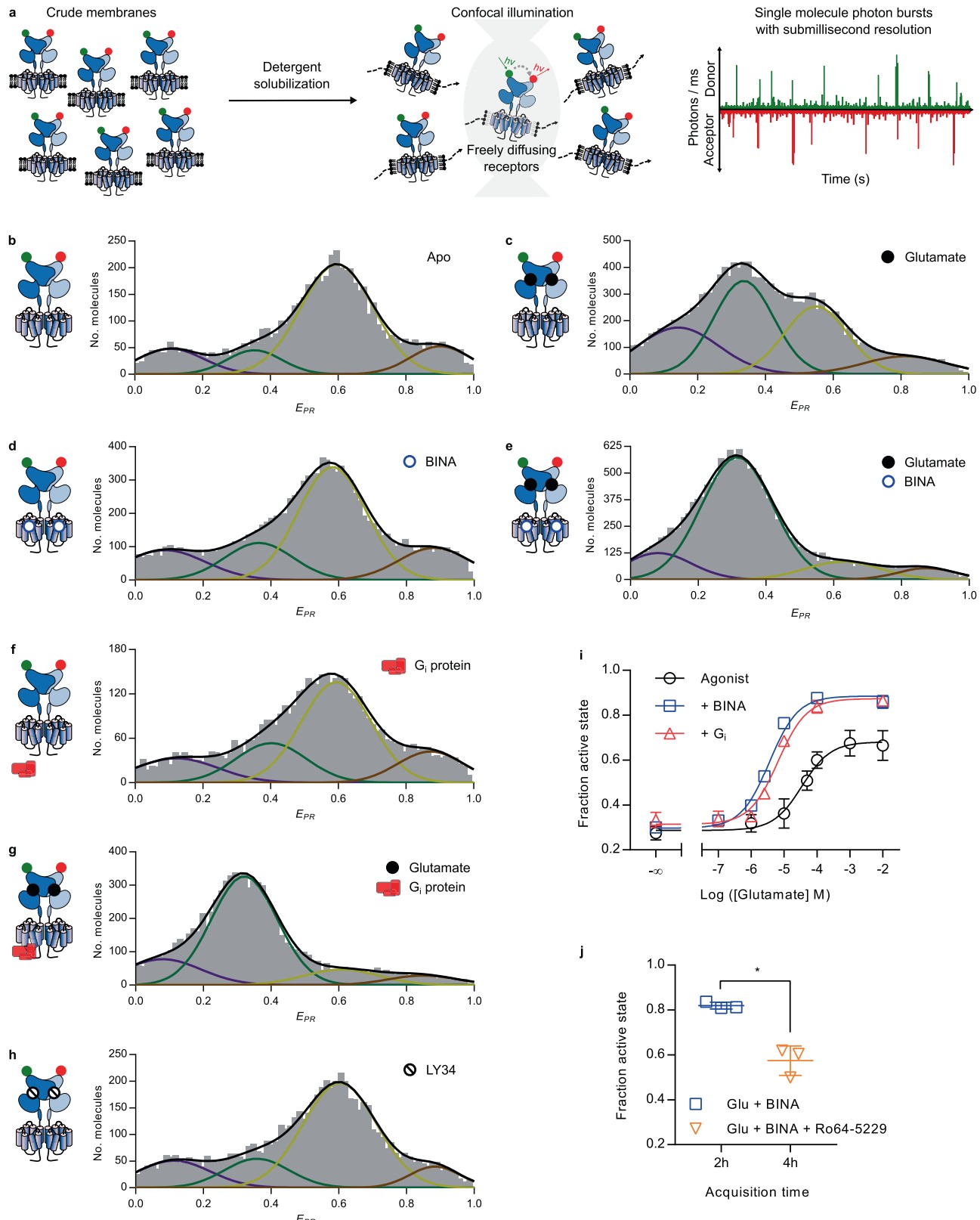

of the major population to lower FRET, a portion of the HF population remained for the full-length receptor.

Next, we explored the effect of a PAM at saturating concentrations. Alone, BINA had no effect (Fig. 3d), which agrees with the expected effect of a pure allosteric modulator that requires an agonist to reveal its modulatory activity. Thus, BINA does not act as an ago-PAM with regard to VFT reorientation. In contrast, in presence of saturating Glu, BINA unveiled its PAM effect and led to a strong increase of the LF population (Fig. 3e), accompanied by a nearly complete depopulation of the HF states.

We then analyzed the influence of the heterotrimeric $G_i$ protein, known to stabilize the high agonist affinity state of other

**Fig. 3 smFRET reveals the conformational landscape of full-length mGlu2 in LMNG-CHS-GDN micelles. a** SNAP-mGlu2 dimers were labeled with cell-impermeable Cy3B donor and d2 acceptor fluorophores on living HEK-293T cells. Then mGlu2 dimers were detergent-solubilized from crude membrane fractions and smFRET measurements were performed on freely diffusing molecules with confocal illumination. **b–h** Representative histograms displaying the number of doubly labeled molecules as a function of apparent FRET efficiency ($E_{PR}$). Distributions were obtained in the absence of ligand (Apo) or in the presence of Glu (10 mM), competitive antagonist LY341495 (1 mM), BINA (10 μM), and G protein (1 μM), as indicated. Colored lines represent Gaussian fitting, black lines correspond to the cumulative fitting (see text). All histograms revealed four major populations at very low FRET (VLF, purple), low FRET (LF, green), high FRET (HF, yellow), and very high FRET (VHF, red). **i** smFRET analysis of the effect of Glu without (Agonist) or with BINA (10 mM) or $G_i$ (1 μM), as indicated. The fraction of the active state is defined as the fraction of molecules in the LF population over all molecules in the LF +HF populations. **j)** smFRET analysis of the reversibility of the PAM-induced full ECD activation (500 nM, 2 h) through competition with an excess of the NAM Ro64-5229 (10 μM, 4 h). The statistical difference was determined using a two-sided, paired $t$ test and is given as $*p = 0.017$. $n = 3$ independent biological replicates examined over 3 independent experiments. **i–j** Data were obtained from three biological replicates and are given with mean ± SD. Source data of panels **b–j** are provided as a source data file.

GPCRs and their fully active conformations[38]. For mGlu2, it is established that the $G_i$ protein influences the VFT reorientation[30]. Interestingly, the addition of the nucleotide-free $G_i$ at saturating concentrations led to nearly identical FRET distributions as those promoted by BINA, both in the absence and presence of Glu (Fig. 3f and g, respectively).

The combination of BINA and $G_i$ did not result in a further detectable synergistic effect (Fig. S16a). We therefore concluded that BINA as well as $G_i$ exert an allosteric control through the 7TM, which is required for a complete reorientation of the VFTs toward the LF state.

Finally, we noted that application of saturating concentrations of the competitive orthosteric antagonist LY341495 (LY34) led to a similar distribution as seen for the apo receptor (Fig. 3h), consistent with a receptor remaining in the resting state[30]. The antagonist was able to bind and properly reverse the effect of a sub-saturating concentration of glutamate (100 μM, Fig. S17). This indicates that no basal receptor activity or residual Glu was observed in our preparations, which was further verified by titration with LY34 in LRET measurements (Fig. S13a–e).

We fitted all distributions with four Gaussians and recovered similar values of $E_{PR}$ and full width half maximum (FWHM), pointing to the fact that similar FRET states are populated for all conditions tested (Fig. S18a–b). The major changes in response to ligands were found to result from the depopulation of the HF state accompanied by an increase of the LF state. Therefore, to gain a quantitative view of mGlu2 receptor activation, we calculated the fraction of active molecules, defined as the fractional amplitude of the molecules found in the LF state relative to the HF + LF states. As no notable changes were observed in the two minor populations at very low FRET (VLF, $E_{PR} \sim 0.1$, purple) and very high FRET (VHF, $E_{PR} \sim 0.87$, red, Fig. S18c–d), these were not included in the analysis. We nevertheless verified that calculating the fraction of the active molecules as VLF + LF relative to all molecules led to similar results.

The fraction of active molecules recovered from the fit of the data obtained as a function of the Glu concentration (Fig. S19) allowed us to plot dose-response curves. $pEC_{50}$ values obtained for Glu in the absence or the presence of saturating concentrations of BINA (Fig. 3i, black and blue curves, respectively) were in good agreement with those obtained from ensemble LRET on membranes (Fig. 2b) or in optimized detergent micelles (Fig. 2g). The allosteric effect of BINA on the apparent Glu potency (an increase by almost one order of magnitude) as well as its effect on the maximum efficacy were also recovered (Fig. 3i). This effect was reversible, as the addition of an excess of the NAM Ro64 to receptors after activation by 500 nM BINA + Glu decreased the fraction of active receptor to a similar level observed when only Glu + NAM were applied (Fig. 5i–j and S16b), being slightly below that observed in the presence of Glu alone (Fig. 3i). Altogether, these results further validated the full functional

integrity and native-like ligand responsiveness of our receptor preparations in optimized detergent micelles.

In addition, Glu titration at saturating $G_i$ concentration was strikingly similar to the one obtained with BINA (Fig. 3i, compare red and blue curves). Thus, $G_i$ acts as an allosteric modulator on Glu potency and VFT activation. Most notably, no additional populations or substantial changes in the four major peak positions ($E_{PR}$) were found in the presence of BINA or $G_i$. This indicates that even if BINA and $G_i$ promote alternative conformations through distinct interaction sites at the 7TM level, their allosteric effect on the VFT conformation can be explained by a simple shift of the equilibrium toward the active state, rather than the stabilization of alternative states.

**BINA or $G_i$ are required to suppress submillisecond dynamics and stabilize the active state.** We then took advantage of the high time resolution of our PIE/nsALEX approach to uncover hidden states, sampled by the receptor during its residence time in the confocal illumination volume (here ~5 ms). Interconversions between multiple FRET efficiency states at timescale faster than this residence time lead to averaging, which results in populations being found at intermediate FRET efficiency values when calculated by integrating over the entire residence time.

We employed two different methods to gain insights into the dynamic behavior of the mGlu2 VFTs in full-length receptors. First, we plotted donor fluorescence lifetimes $\tau_{DA}$ for each single molecule against the γ-corrected FRET efficiency $E$ ("$\tau_{DA}$ vs. $E$" analysis[39]). This representation allows to identify structural dynamics, if populations deviate from the theoretical "static FRET line" (yellow line, Fig. 4a–c and S20). For apo receptors, the main HF population appeared above the static FRET line (Fig. 4a), thus indicating submillisecond conformational oscillations. In contrast, the LF population promoted by application of Glu (Fig. 4b) and further populated in the presence of Glu + BINA (Fig. 4c) was found much closer to the static FRET line, therefore implying reduced dynamics of the active VFT state.

Second, we performed time windows analysis (TWA)[40], which relies on recalculating the FRET efficiency at integration times shorter than the residence time, here from 1 ms down to 200 μs. Shortening the integration time below 300 μs strikingly led to the disappearance of the main HF population for apo receptors, while two populations at $E \sim 0.2$ and >0.9 were revealed (Fig. 4d, red). This indicates that at integration times longer than 300 μs the apparent FRET population centered at $E \sim 0.6$ represents the time-averaged FRET value between these two sampled states. We therefore conclude that at sub-millisecond timescales, the apo receptor samples a set of conformations at low and very high FRET values, representing the active and inactive states, respectively (Fig. 4g). Of note, the distribution obtained by integration at 1 ms (Fig. 4d, green) matched the one obtained

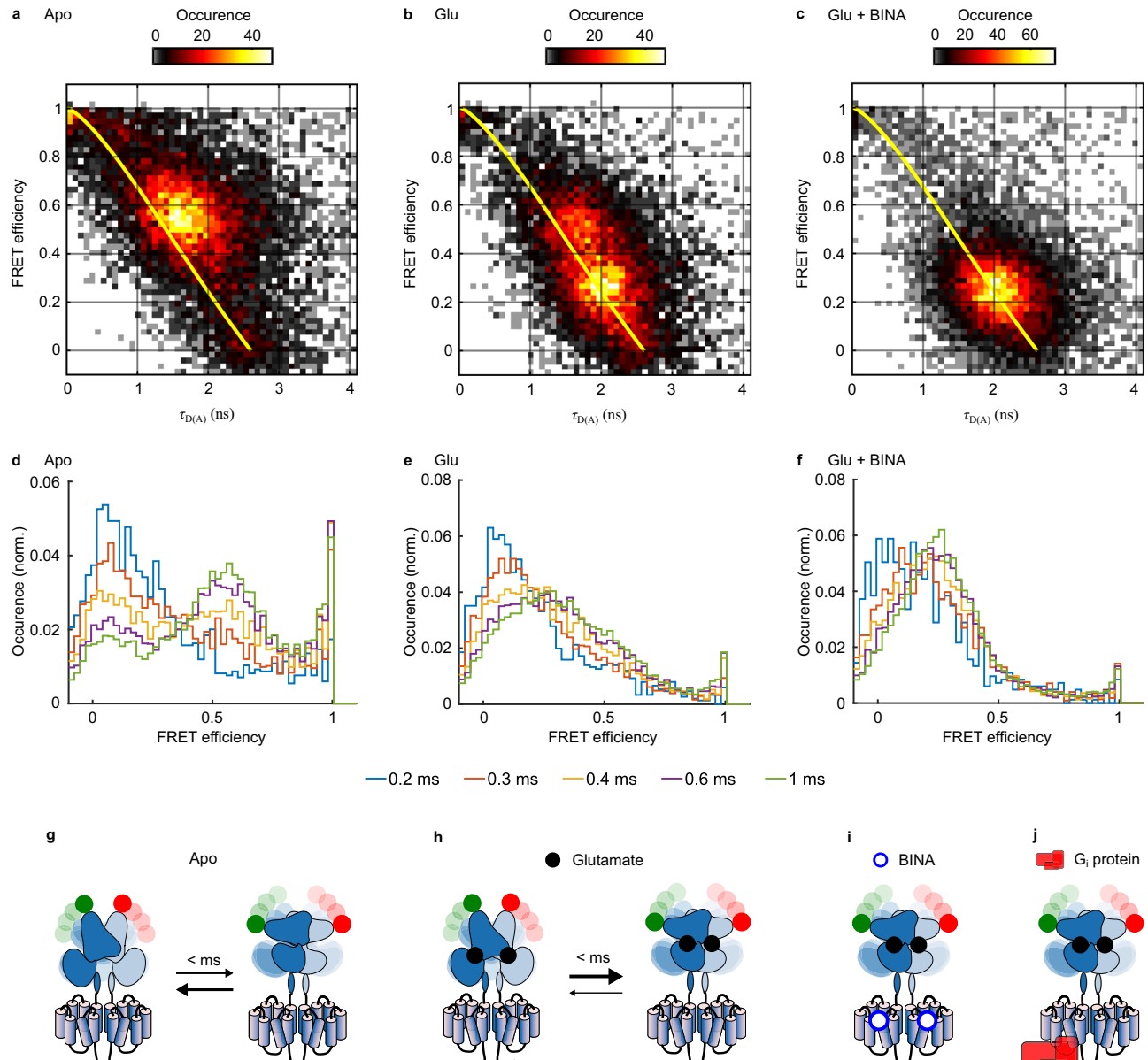

**Fig. 4 Structural dynamics analysis of mGlu2 dimers in response to orthosteric and allosteric ligands. a–c** Representative $\tau_{DA}$ vs. E histogram for mGlu2 dimers in the absence (Apo) or presence of Glu or Glu + BINA. For the Apo receptors, the major population deviates from the "Static FRET" line (yellow), indicating conformational dynamics at the submillisecond time scale. The addition of Glu stabilizes the VFT in an ensemble of low FRET conformations with less flexibility, an effect that is reinforced by the allosteric modulator BINA. **d–f** Time window analysis for different integration times (from 0.2–1 ms) reveals large conformational flexibility of the Apo VFT at 200–600 μs timescales, which is strongly restricted when bound to orthosteric agonist and allosteric modulator. The number of data points used varied from 2183-1640-901 (**d–f** respectively, 200 μs) to 14082-13132-13269 (1 ms) **g–j** Schematic representation of the major species observed in all cases, with the timescales of the transition between them. Black and blue dots represent Glu and BINA, respectively. Source data of panels **a–f** are provided upon request.

from calculations integrated over the entire residence time (Fig. 3b), indicating no detectable dynamics between 1 ms and the residence time of ~5 ms. The addition of the antagonist LY34 led to a distribution similar to the apo state (Figs. S21a and 22a), thus excluding a stabilization of the oscillating VFTs by this orthosteric ligand.

In contrast, orthosteric and allosteric ligands promoted stabilization of the VFTs in an ensemble of low FRET conformations (Fig. 4e–f). This was particularly obvious in the presence of Glu + BINA, where the majority of molecules remained within the LF population at $E \sim 0.25$ even at lower integration times. Indeed, only some residual conformational

dynamics, limited to low FRET states, were observed at an integration time of 200 μs (Fig. 4f). A similar stabilization was observed for Glu-activated receptors in the presence of $G_i$, underlining the close similarity of allosteric modulation exerted by PAM and G protein on VFT dynamics (Figs. S21b and 22b). Altogether we concluded that the synergistic binding of Glu and a positive allosteric modulator, either the G protein or a small synthetic PAM, promoted the stabilization of the mGlu2 receptor in an ensemble of conformations characteristic of the active state, stable for at least several milliseconds as given by the residence time in the confocal volume (Fig. 4i-j). In the case of receptors bound to Glu alone, this stabilizing effect of the active state was

less pronounced (Fig. 4b and e), with a fraction of receptors still sampling between the high FRET resting state and the low FRET active state (Fig. 4h).

Application of the NAM Ro64 in the presence of Glu did not alter these partially remaining dynamic behavior (Fig. S21c and 22c). Overall, this confirms the inability of the natural full agonist Glu to fully stabilize the VFTs in the active state and emphasizes the importance of a long-range functional link between the 7TMs and the ECDs that allows for allosteric interdomain communications mandatory for maximal stabilization of the VFTs in the active state.

**Loss of functional integrity and dynamics of mGlu2 in IGE-PAL.** Our LRET data demonstrated that human mGlu2 receptors only provide a strong and stable response to PAM in LMNG-CHS-GDN micelles, while in IGEPAL or DDM micelles, even supplemented with CHS, this effect was completely absent or low and unstable (Fig. 2, S1–3). A similar observation was made for another SNAP-sensor based on the rat receptor (Fig. S23) that was used in previous LRET and single-molecule studies[25,26,30]. To further understand the differential effects of these detergent mixtures, we analyzed the effect of Glu, BINA, $G_i$, and Ro64 on receptors solubilized with IGEPAL by smFRET (Fig. S24). In contrast to the data obtained in LMNG-CHS-GDN, we found that Glu was sufficient to totally stabilize the receptors in the active VFT state (Fig. S24a–c, S21d–e, and S22d–e), similarly to earlier reports[25]. No further effect on VFT activation was observed upon the addition of BINA or the $G_i$ protein (Fig. S24a and d–e, respectively) and likewise the negative allosteric modulator Ro64 was not capable of reducing the fraction of the active state (Fig. S24a and f) in a way seen in LMNG-CHS-GDN. These observations underline the loss of allosteric effects in IGEPAL and likely other detergents mixtures. Such lack of functionality of the receptor could arise from: 1/ a loss of allosteric communication between the 7TM and the ECD; 2/ a loss of structural integrity of the 7TM that becomes unable to bind the PAM, NAM, and G protein, or 3/ a direct effect of IGEPAL on the conformation of the 7TM, stabilizing it in a PAM-bound-like confirmation, which should only be reached in the presence of the allosteric modulator under native-like conditions.

**Maximal VFT activation remains ligand-dependent.** Next, we addressed the mode of action of partial agonists, previously shown to promote changes in the VFT intersubunit orientation but to a lower extent than full agonists[30]. Pharmacologically, partial agonists are ligands that do not trigger maximal cellular responses, not even at saturating concentrations[41]. At the structural level, this may either be explained by the existence of specific intermediate active conformations[42] or by a less efficient shift of the resting-to-active equilibrium compared to full agonists. Our previous data proposed a simple shift in the equilibrium of isolated VFTs dimers rapidly oscillating between active and resting conformations toward the active state, while maintaining submillisecond dynamics[31]. Here, in full-length receptors in LMNG-CHS-GDN, the $E_{PR}$ peak positions of the four populations described in Fig. 3 were perfectly recovered for the partial agonists LCCG-I (Fig. 5a), DCG-IV (Fig. 5b) and LY354740 (LY35) (Fig. S16c). Nevertheless, the extent of depopulation of the HF state and the corresponding increase in the population of the LF state remained ligand-dependent. Quantification of the fraction of activation indicated that these molecules have lower efficacy than Glu to populate the active state (Fig. 5i–j). We further observed submillisecond dynamics of the HF state under these conditions, pointing to the inability of these partial agonists to efficiently stabilize the less dynamic active VFT state. Indeed, the HF

population appeared above the static FRET line (Fig. S21f–h), while the FRET distributions in TW analysis remained intermediate between those of the apo and the Glu-bound receptors (Fig. S22f–h). The addition of BINA (Fig. 5c–d) or $G_i$ (Fig. 5e–f) further pushed the equilibrium toward the active state, but to a lower extent than obtained with Glu (Fig. 5i–j). This observation revealed that these partial agonists are unable to fully stabilize the VFT in the active orientation, even in the presence of BINA or the heterotrimeric $G_i$ and consequently, that maximal VFT activation still remains dependent on the individual efficacy of an agonist. Furthermore, these results together with the finding that all studied conditions resulted in the same four major FRET states (Fig. S18a), point to a model where that partial agonists do not stabilize intermediate FRET states but shift the equilibrium between the dynamic inactive and the less dynamic active VFT states.

**The natural full agonist glutamate does not exhibit maximal efficacy.** Finally, we further characterized the synthetic full agonist LY379268 (LY37) at the single-molecule level. Interestingly, this ligand appeared more potent than Glu to stabilize the VFT in its active state (Fig. 5g, i–j). The $E_{PR}$ histogram showed a higher fraction of molecules in the active state than for Glu. Similarly, the dynamic analysis showed a stabilization of the majority of molecules in the LF states for up to several milliseconds (Fig. S21i–22i). This observation points to the possibility that LY37 might qualify as a superagonist[43], i.e. a compound that displays greater efficacy and thus higher receptor signaling output, than the endogenous full agonist Glu. However, this effect is only observed when the receptor is solely bound by the orthosteric ligand, as the distribution of states obtained upon activation in the presence of PAM was identical for receptors bound by Glu and LY37 (Fig. 5g–j).

**Discussion**
GPCR activation can be finely tuned by different classes of ligands acting either via orthosteric or allosteric sites. Among them, PAMs enhance agonist action by increasing their potency and/or efficacy. Here, we used smFRET to explore how a PAM can increase the efficacy of mGlu2 receptors, by monitoring the fast dynamics of the intersubunit rearrangement of the VFTs. We analyzed the effect of BINA, a mGlu2 specific PAM, on isolated, full-length, and fully functional receptors with submillisecond time resolution, relevant for the conformational movements of such protein domains[44]. Our data reveal the presence of four VFT states. Two of them - the HF/inactive and LF/active states - are predominantly populated in a ligand-dependent manner (Figs. 3, 5 and S16), Based on previous studies[25,30], we attributed the LF state and the HF states to conformations in the "active" and "resting" orientations of the dimeric ECD respectively, with both VFTs in the "closed" and "open" conformations (Acc and Roo), respectively. Two minor populations (VHF and VLF) were barely affected by ligands (Fig. S18a and b, respectively), but we note that they could be in exchange with the two major populations at timescales slower than the resolution of our method (> 5 ms). Interestingly, in a very recent study using a construct highly similar to ours, the structural dynamics of mGlu2 were monitored using smFRET at the surface of living cells, with a lower time resolution (40 ms)[45]. The transition from a high FRET ($E = 0.44$) to a low FRET ($E = 0.3$) state were observed upon glutamate binding. But notably, some transitions to a very high FRET state ($E = 0.88$) and to a lower FRET state (not reported by the authors) could be observed in some traces. We hypothesize that these seldom populated VHF and VLF states could correspond to

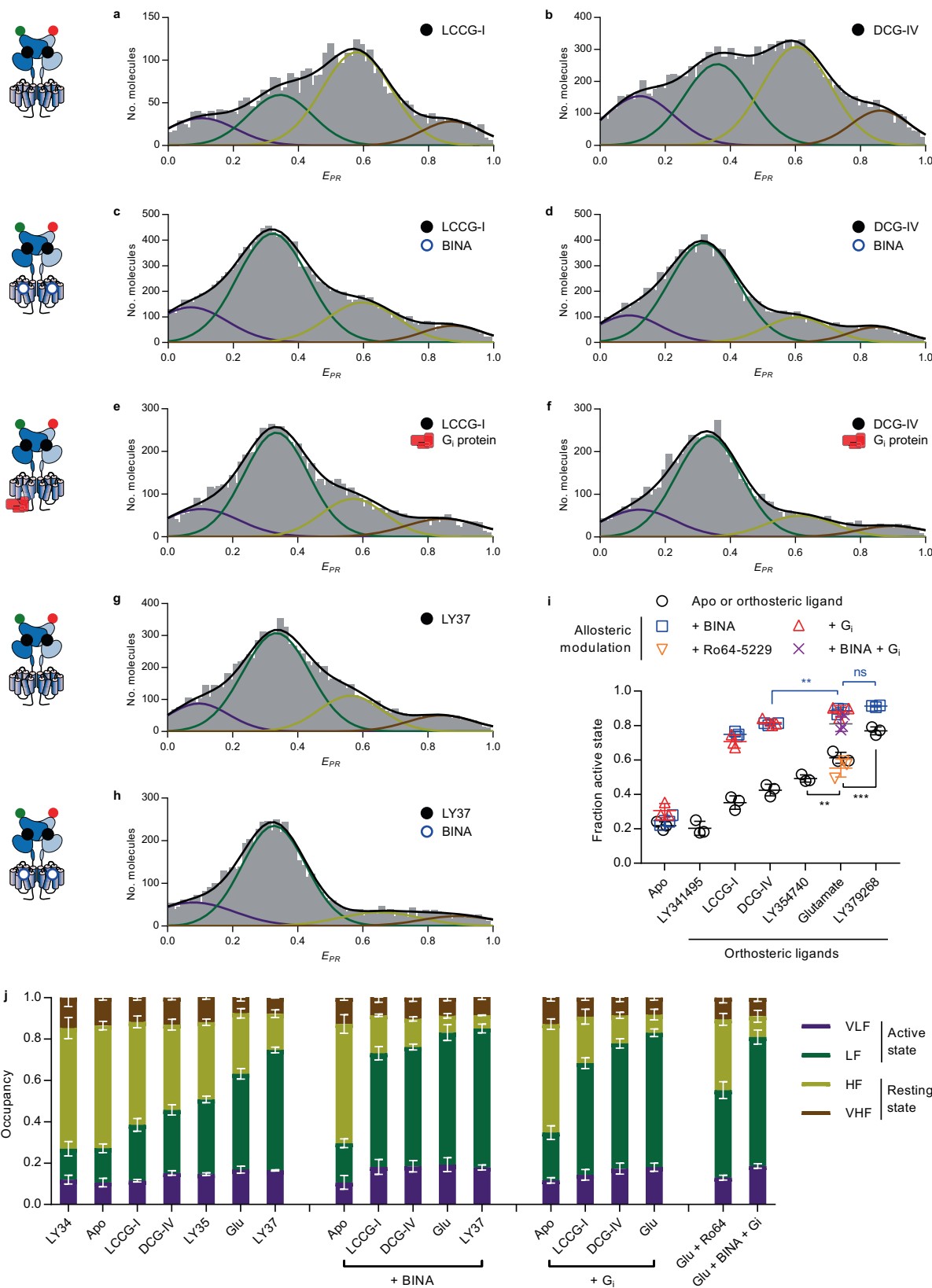

the ones observed in our study, although the determination of their exact structural nature would require further studies.

The conformational landscape of receptor populations clearly differed from the one observed with the isolated VFT dimers[31]. In that latter case, all dimers were shown to be oscillating at a ~100 μs timescale between high and low FRET states, in response to all ligands tested. Here, in the case of full-length receptor dimers in the apo state or bound to antagonist, the main population is similarly oscillating between the HF and LF states (at a slightly slower timescale of ~200–500 μs, Fig. 4 and S20–22). However, in contrast to the isolated VFT dimer, the addition of full agonist led to a stabilization of an ensemble of LF/active

**Fig. 5 Different efficacies of orthosteric ligands on mGlu2 ECD rearrangement. a–h** FRET distributions were obtained in the presence of partial agonists LCCG-I (**a**) and DCG-IV (**b**) or full synthetic agonist LY379268 (**g**) alone or in the presence of BINA (**c**, **d**, **h** respectively) or G$_i$ (**e** and **f**, respectively). **i** Comparison of the fraction of the active state (LF/(LF+HF)) in response to different orthosteric and allosteric ligands. $n = 3$ independent biological replicates examined over 3 independent experiments for each condition. The scatter plot shows data together with the mean± SD. Statistical differences were determined using a one-way ANOVA with Sidak multiple comparisons test and are given as: $p_{LY35/Glu} = 0.0068$ (**), $p_{Glu/LY37} = 0.0006$ (***), $p_{DCG-IV+BINA/Glu+BINA} = 0.0028$ (**), $p_{Glu+BINA/LY37+BINA} = 0.61$ (ns). **j** Comparison of the fraction of all states in response to different orthosteric and allosteric ligands. Data represent the stacked means ± SD from 3 independent biological replicates for each state with error bars centered around the mean for the VLF, LF, HF, and VHF from top to bottom, respectively. Source data of panels **a–j** are provided as a source data file.

states, an effect further promoted by a PAM. These states appear stable for at least several milliseconds, a duration compatible with the activation of downstream signaling[46,47]. We propose that this stabilization of the active VFT state stems from a strengthening of the active dimeric interface, probably via interactions involving transmembrane helix 6, as reported based on crosslinking experiments at the surface of live cells[48,49] and structure determination for mGlu5[16], mGlu1[17] and GABA$_B$ receptors (Shaye et al. 2020). However, one cannot exclude that the T7M bundles through their contact with the CRDs, also allosterically stabilize the CRD and VFT dimer in their active orientation.

Saturating concentrations of partial agonists or of the natural agonist Glu were not able to fully depopulate the basal HF state and stabilize receptors in the active LF state during the observation time of several milliseconds (Figs. 3, 5, and S20-22). The addition of BINA to the partial agonists was not sufficient to promote the stabilization of the active state to the extent observed with the full agonists Glu and LY37 (Fig. 5i–j). Thus, the extent of activation remains ligand-dependent even in the presence of allosteric modulators. In contrast, LY37, formally considered a full agonist like Glu, appears more efficient than Glu in promoting the active VFT state in our assay (Fig. 5i–j), which qualifies this molecule as a "superagonist"[43]. Likely, this effect was previously hidden in cell-based assays, due to amplification of the signaling cascade and saturation of the readout[30].

The presence of the nucleotide-free heterotrimeric G$_i$ protein complex was found to produce the same effect as the PAM, allowing Glu to fully stabilize the active state (Fig. 3). Like BINA, G$_i$ also increased partial agonist efficacy in populating the active state (Fig. 5). Of note, the effects of BINA and G protein are not additive, suggesting they exert a similar effect. This is consistent with our observation that the G protein-bound state of another class C GPCR (the GABA$_B$ receptor), is similar to that observed with an agonist and a PAM[8,50]. Our data are also consistent with the positive allosteric action of G proteins on GPCRs[38], as also observed with class C receptors, including mGlu2[30]. Within the cellular environment, such an effect is expected to be transient, as upon GTP binding, the receptor-G protein complex dissociates and the allosteric action is lost. As such, PAMs can maintain this effect by stabilizing receptors in their active G protein-bound-like state, which then facilitates G protein binding and activation.

Our data contradict those previously reported, showing an apparent full stabilization of the mGlu2 receptor in its active state with Glu alone in IGEPAL conditions[25–27]. It is possible that the lower time resolution in these assays prevents the detection of the basal state population. However, our results obtained in these conditions of detergents (Fig. S24) suggest that the 7TM domains behaves as if they were already occupied by a PAM, likely already being in an active-like state. Interestingly, a recent study using smFRET on a sensor reporting on the proximity between the CRD domains suggested that the fully active conformation of the receptor could not be reached in the presence of Glu alone[51]. Stabilization in the fully active state required the C770A mutation in the 7TM domain, described to enhance mGlu signaling in a manner similar to a PAM. Although this observation supports

our model that a PAM effect is required for the full stabilization of the mGlu2 active state, we note that the effect of the binding of a synthetic PAM or of the G$_i$ protein was not described in that study. It is likely that the receptor was in fact not able to be activated by such PAM under the detergent conditions used (DDM (0.05%) + CHS (0.005%)), for which we report here the absence of effect of BINA under very similar detergent conditions (Fig. S3 and 23).

Therefore, our results and the comparison with previous studies demonstrate that careful optimization of solubilization conditions is required to maintain the functional integrity of full-length mGlu2 at room temperature. This was only achieved using a mixture of LMNG-CHS-GDN, while all other tested conditions employing popular detergents exhibited a time-dependent impact on receptor function (Fig. 3h). It is not surprising that a functional reconstitution of the multidomain, multimeric mGlu receptor requires adapted characteristics to account for proper folding, ligand binding, and activity. Improved functionality by the branched nonionic detergent LMNG through enhanced stabilization of the 7TM[52] and beneficial polar interactions of the maltoside head with loops and 7TM ends may play an important role in maintaining the functional link between the ECD and the 7TM[16]. Further stabilization and functionality are provided by the two sterol-containing compounds CHS and GDN. Both mimic cholesterol, known to be important for class C GPCR function[19,53] and CHS further provides a net negative charge to the detergent micelles. Negative charges have been described to enhance agonist affinity and stabilize the active state of the β2-AR[54], a prototypical class A GPCR, whose orthosteric binding site comprises similar features to that of the allosteric site in mGlu[55]. Nevertheless, the triple combination LMNG-GDN-CHS was required to maintain receptor function over time, pointing to a complementary role of these molecules. Taken together, these observations highlight the importance of the lipid environment on mGlu receptor function.

Overall, by identifying conditions under which the solubilized mGlu2 receptor conserves its modulation by a PAM and the G protein, we have been able to show that BINA can increase the population of active receptors in the presence of Glu. Our data point to a model where the increased efficacy observed with this PAM would arise from its ability to stabilize the active state already populated in the presence of orthosteric agonists. However, the validation of such a model would require a deeper investigation of the conformations sampled by mGlu2 by measuring distances and distances changes between its various structural modules, using the incorporation and the labeling of unnatural amino acids for example[51,56]. Such studies will pave the way for a deeper understanding of how the structural dynamics of mGlu receptors as well as other membrane receptors regulate their function and may open up alternative routes for the development of fine-tuned therapeutics.

## Methods

**Chemicals**. All chemicals were purchased from Sigma-Aldrich, Merck and Roth unless otherwise noted. n-dodecyl-β-D-maltopyranoside (DDM), lauryl maltose

neopentyl glycol (LMNG), and cholesteryl hemisuccinate (CHS) tris salt were purchased from Anatrace (through CliniSciences, France). Glyco-diosgenin (GDN) was purchased from Avanti Polar Lipids through Merck. SNAP-Lumi4-Tb, SNAP-green, SNAP-Cy3B, and SNAP-d2 were obtained from Cisbio Bioassays (Codolet, France). DCG-IV, LY341495, LY379268, LY354740, LCCG-I, BINA hydrochloride, and Ro64-5229 were purchased from Tocris Bioscience (Bristol, UK).

**Plasmids**. The pcDNA plasmid encoding SNAP-tagged human mGlu2 was a gift from Cisbio Bioassays (Codolet, France) and is described in Fig. S25.

**Cell culture**. Adherent HEK293T cells (ATCC CRL-3216, LGC Standards S.a.r.l., France) were cultured in Gibco™ DMEM, high glucose, GlutaMAX™ Supplement, pyruvate (Thermo Fisher Scientific, France) supplemented with 10% (vol/vol) FBS (Sigma-Aldrich, France) at 37 °C, 5% $CO_2$ and passaged twice per week.

**Transfection and labeling**. $1 \times 10^7$ cells were seeded in 75 cm² flasks 24 h prior to transfection with Polyethylenimine (PEI 25 K, Polysciences Europe GmbH, Germany) at a DNA to PEI ratio (w/w) of 1:3 using 12 μg mGlu2 plasmid DNA per flask. In brief, 10 mg/ml PEI stock solution in 1 M HCl was diluted in 20 mM MES at pH 5 with 150 mM NaCl and incubated at room temperature for 25 min before sequential addition of 2.5 ml complete medium followed an additional 7.5 ml. The flask culture medium was than replaced by the diluted transfection mix and protein expression proceeded for 48 h at 37 °C, 5% $CO_2$.

SNAP-tag labeling was performed on surface-adhered cells in DMEM GlutaMax without FBS for 1–2 h at 37 °C using final concentrations of either 100 nM SNAP-Lumi4-Tb and 60 nM SNAP-green for LRET or 600 nM SNAP-Cy3b and 300 nM SNAP-d2 for smFRET measurements. Following labeling, excess dye was removed by three cycles of washing with DPBS w/o $Ca^{2+}$ and $Mg^{2+}$ (Thermo Fischer Scientific, France) at ambient temperature.

**Preparation of crude membrane fractions**. Adherent cells were detached mechanically using a cell scraper in DPBS w/o $Ca^{2+}$ and $Mg^{2+}$ (Thermo Fisher Scientific, France) and collected at 500 × g and 22 °C. Subsequently, cells were resuspended in cold hypotonic lysis buffer (10 mM HEPES pH7.4, cOmplete™ protease inhibitor), frozen, and stored at −80 °C. After thawing, cells were passed through a 0.4 mm gauge needle 30-times using a syringe on ice. After two rounds of centrifugation at 500 × g and 4 °C for 5 min, the supernatant was aliquoted and centrifuged at 21,000 × g and 4 °C for 30 min to collect crude membranes. The pellets were washed once with 20 mM HEPES pH 7.4, 118 mM NaCl, flash-frozen in liquid $N_2$, and stored at −80 °C.

**Detergent solubilization**. Receptors were solubilized on ice by resuspension of crude membranes in acquisition buffer (20 mM Tris-HCl pH7.4, 118 mM NaCl, 1.2 mM $KH_2PO_4$, 1.2 mM $MgSO_4$, 4.7 mM KCl, 1.8 mM $CaCl_2$) supplemented with 1% (v/v) IGEPAL, 1% (w/v) DDM, 1% (w/v) DDM + 0.2% (w/v) CHS Tris, 0.1% (w/v) LMNG, 0.1% (w/v) LMNG + 0.1% (w/v) GDN, 0.1% (w/v) LMNG + 0.004%, 0.008% or 0.016% (w/v) CHS Tris or 0.1% (w/v) LMNG + 0.008% (w/v) CHS Tris + 0.05%, 0.1% or 0.2% (w/v) GDN. After 5 min, the solution was centrifuged for 10 min at 21,000 × g and 4 °C. The supernatant was then applied to a Zeba Spin Desalting Column (7 kDa cut-off, Thermo Fisher Scientific, France) equilibrated in acquisition buffer containing 5% of the detergent concentration used for solubilization and centrifuged 2 min at 1500 × g and 4 °C. The flow-through was then immediately diluted 1:20 in cold acquisition buffer and kept on ice in the dark until use.

**LRET**. Intersubunit LRET measurements of mGlu2 dimers, N-terminally labeled with the Lumi4-Tb donor and the green acceptor via an engineered SNAP-tag, were performed on a PHERAstar FS microplate reader (BMG Labtech, Germany) in white 384 well plates (polystyrene, flat-bottom, small volume, medium-binding, Greiner Bio-One SAS, France). Measurements where performed in acquisition buffer in the presence of indicated ligands at room temperature and plates where sealed and stored in the dark in-between measurements for time-course experiments to minimize evaporation and fluorophore bleaching. The fluorescence decay of donor and acceptor was recoded using the LRET 337/620/520 optical module by excitation with 20 flashes per well every 5 μs for a total of 2500 μs. The FRET signal was expressed as sensitized acceptor emission integrated between 50–100 μs and normalized to its emission between 1200–1600 μs.

**Expression and purification of heterotrimeric $G_{i1}$**. The heterotrimeric $G_{i1}$ complex was a kind gift from Sebastien Granier and Remy Sounier (IGF Montpellier, France). $G_{i1}$ heterotrimer was expressed in Sf9 insect cells in EX-CELL 420 media (Sigma). Human $G_{\alpha i1}$ was cloned into the pVL1392 vector, and the virus was prepared using the BestBac system (Expression Systems, LLC). N-terminal Flag-tagged human $G_{\beta 1}$, and human $G_{\gamma 2}$ were cloned into the pFastBac vector, and the virus was prepared using the Bac-to-Bac baculovirus system. The cells were infected with both $G_{\alpha i1}$ and $G\beta\gamma$ virus at a ratio determined by small-scale titration experiments at 27 °C for 48 h before collection. Cells containing $G_{i1}$ heterotrimer were lysed in hypotonic buffer containing 10 mM Tris pH 7.4, 100 mM $MgCl_2$, 10 mM GDP, 5 mM β-mercaptoethanol, and protease inhibitors. After

centrifugation, membranes were dounced and solubilized in a buffer containing 20 mM HEPES pH7.5, 100 mM NaCl, 1% DDM, 5 mM $MgCl_2$, 10 mM GDP, 5 mM β-mercaptoethanol, and protease inhibitors. The solution containing the $G_{i1}$ heterotrimeric complex was loaded onto an anti-FLAG M1 affinity column. After washing of the column with 5 column volumes of buffer E1 (20 mM HEPES pH 7.5, 100 mM NaCl, 1% DDM, 5 mM $MgCl_2$, 10 mM GDP, 5 mM β-mercaptoethanol) and buffer E2 (20 mM HEPES pH 7.5, 50 mM NaCl, 0.1% DDM, 1 mM $MgCl_2$, 10 mM GDP, 100 μM TCEP) at a flow rate of 2 ml min⁻¹. After a detergent exchange was performed by washing the column with a series of seven buffers (3 CV each) made up of the following ratios (v/v) of LMNG buffer (20 mM HEPES pH 7.5, 50 mM NaCl, 0.5% LMNG, 1 mM $MgCl_2$, 10 mM GDP, 100 μM TCEP) and E2 buffer: 0:1, 1:1, 4:1, 9:1, 19:1, 99:1 and LMNG buffer alone. $G_{i1}$ heterotrimer was eluted with Elution buffer (20 mM HEPES pH 7.5, 50 mM NaCl, 0.01% LMNG, 1 mM $MgCl_2$, 10 mM GDP, 100 μM TCEP). The eluted sample was concentrated in a 50 kDa MWCO concentrator to 100 μM and aliquots were flash-frozen in liquid Nitrogen and stored at −80 °C.

**PIE-MFD smFRET setup**. Single-molecule FRET experiments with pulsed interleaved excitation (PIE) – multiparameter fluorescence detection (MFD) were performed on a homebuilt confocal microscope[56] using the SPCM 9.85 software (B&H)

In brief, the 20 MHz-pulsed white excitation laser was split into two beams spectrally filtered using excitation bandpass filters at wavelength 532/10 (prompt beam) and 635/10 (delayed beam) to excite the Cy3b donor and d2 acceptor molecules, respectively. The delayed beam has a path length of ~8 m relative to the prompt beam, generating a ~24 ns delay in the pulse. The two beam paths are then recombined and focused using a 10× objective into a single-mode fiber, by which the beams become spatially overlapped and filtered. The output of the fiber is collimated using a 10× microscope objective lens, polarized, and coupled into an inverted microscope (Eclipse Ti, Nikon, France). The excitation power was controlled to give 25 μW for the prompt and 12 μW for the delayed beam at the entrance into the microscope. Inside the microscope, the light is reflected by a dichroic mirror that matches the excitation/emission wavelengths (FF545/650-Di01, Semrock, Rochester, NY, USA) and coupled into a 100 x, NA1.4 objective (Nikon, France). Emitted photons are then collected by the objective and focused on a pinhole of 150 μm. The emission photon stream is collimated and divided using a polarizing beamsplitter. In each created polarization channel, the photons are spectrally separated using dichroic mirrors (BS 649, Semrock, Rochester, NY, USA) and filtered using high-quality emission filters (parallel: ET BP 585/65, Chroma, Bellows Falls, VT, USA, and FF01-698/70-25, Semrock, Rochester, NY, USA, perpendicular: HQ 590/75 M, Chroma, Bellows Falls, VT, USA and FF01-698/70-25, Semrock, Rochester, NY, USA). Single photons are detected using Single Photon Avalanche Diodes. We use two MPD-1CTC (MPD, Bolzano, Italy) for the donor wavelength channels and two SPCM AQR-14 (Perkin Elmer, Fremont, CA, USA) for the acceptor wavelength channels. The output of the detectors is coupled into a TCSPC counting board (SPC-150, Becker&Hickl, Berlin, Germany), through an HRT41 router (B&H), using appropriate pulse inverters and attenuators. The sync signal that triggers the TCSPC board is provided by picking a small fraction of the light from the prompt path (reflected by a coverslip), and focusing it on an avalanche diode (APM-400, B&H).

**smFRET measurements**. Measurements were performed at concentrations of 30–100 pM on SensoPlate 384 well plates (non-treated, Greiner Bio-One, France) passivated with 1 mg/ml bovine serum albumin (BSA) in acquisition buffer with detergent for at least 1 h prior to sampling application. Samples were measured in acquisition buffer (20 mM Tris-HCl pH 7.4, 118 mM NaCl, 1.2 mM $KH_2PO_4$, 1.2 mM $MgSO_4$, 4.7 mM KCl, 1.8 mM $CaCl_2$) with detergent and ligand concentrations as indicated in the text and in the absence of any oxygen scavenging system or triplet state quenchers. Measurements at saturating ligand concentration were carried out at 10 mM Glu, 100 μM LY37, 100 μM LY34, 1 mM LCCG-I, 1 mM DCG-IV, and 1 mM LY35. Allosteric modulators BINA and Ro64 were supplemented at a final concentration of 10 μM. The effect of BINA at 500 nM was reversed by the addition of 10 μM ro64. To study the effect of heterotrimeric human $G_{\alpha i1}G_{\beta 1\gamma 2}$ on ECD reorientation 1 μM of the heterotrimer in the absence or presence of ligand was incubated with the receptor (at approximately 30–100 pM) for 30 min at room temperature in the presence of 1 μM TCEP, 100 μM GDP, followed by the addition of 0.05 U/μl of apyrase (Sigma-Aldrich, France) and incubation for another 30 min before acquisition. The effect of 100 μM Glu on VFT reorientation was reversed by the addition of 1 mM of the competitive antagonist LY34.

**smFRET data analysis**. Data analysis was performed using the Software Package for Multiparameter Fluorescence Spectroscopy, Full Correlation and Multiparameter Fluorescence Imaging developed in C.A.M. Seidel's lab (http://www.mpc.uni-duesseldorf.de/seidel/). A single-molecule event was defined as a burst containing at least 40 photons with a maximum allowed interphoton time of 0.3 ms and a Lee-filter of 20. Photobleaching events were identified base on |TGX-TRR| < 1 ms as described[57].

$\tau_{DA}$ vs $E$ analysis and time windows analysis were performed using the PAM 1.3 software[58]. The static FRET line for the $\tau_{DA}$ vs. $E$ analysis was plotted taking into consideration the excited-state lifetime of the donor, and a 6 Å dye linker length. A minimal threshold of 25 photons per time bins was used in the time windows analysis.

Apparent FRET efficiencies ($E_{PR}$), FRET efficiencies ($E$), and Stoichiometry ($S$) were calculated using the conventions and recommendations made in[59] and [36]

$$E_{PR} = {}^{iii}E_{app} = \frac{F_{A/D}}{F_{A/D} + {}^{ii}F_{Dem/Dex}} \quad (1)$$

$$E = \frac{F_{A/D}}{F_{A/D} + \gamma \cdot {}^{ii}F_{Dem/Dex}} \quad (2)$$

Where,

${}^{ii}F_{Xem/Yex}$ is the background-corrected intensity in the X emission channel upon Y excitation. $F_{A/D}$ are the detected intensities in the acceptor emission channel upon donor-excitation, corrected for background, donor leakage α (fraction of the donor emission into the acceptor detection channel), and direct excitation δ (fraction of the direct excitation of the acceptor by the donor-excitation laser) γ is the normalization factor that considers effective fluorescence quantum yields and detection efficiencies of the acceptor and donor. The values used for these corrections were α = 0.217, δ = 0.095, and γ = 1.18 Note that we did not see any effect of the ligands or the detergents on the fluorophore properties such as excited-state lifetime (as measured for donor and acceptor) or relative brightness/quantum yield (as measured by determination of the γ factor). In LMNG-CHS-GDN minor contaminations of molecules appearing as donor-only with a lifetime $\tau_D > 3$ ns were observed. These molecules were removed from our analysis solely to determine the donor leakage α factor but considered in all further analysis. In the presence of IGEPAL, significant contaminations of molecules with a lifetime $\tau_A > 2$ ns and molecules appearing as donor-only with a lifetime $\tau_D > 3$ ns (Fig. S26) were observed. Molecules with $\tau_D > 3$ ns were removed only for the determination of α while molecules with $\tau_A > 2$ ns were completely removed from our analysis.

To display the 1D FRET efficiency histograms and for further analysis, doubly labeled (Donor-Acceptor) molecules were selected using the stoichiometry $S$ factor using (0.3–0.35 < $S$ < 0.6–0.65).

**Additional software.** The structures shown in Fig. 1 were generated using PYMOL 2.3.3. LRET data was analyzed using MARS (BMG Labtech) and displayed in GraphPad PRISM 7.05. FRET histograms were fitted and displayed using Origin 6 (Microcal Software, Inc.) and PRISM 7.05 (GraphPad). Figures were generated using Microsoft PowerPoint 2019 and INKSCAPE 0.92

**Reporting summary.** Further information on research design is available in the Nature Research Reporting Summary linked to this article.

## Data availability

Source data are provided with this paper. Raw data of smFRET acquisitions will be provided upon reasonable request. References Source data are provided with this paper.

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

## Acknowledgements

We thank Sebastien Granier and Rémy Sounier (IGF, Montpellier, France) for providing the heterotrimeric G protein; Gilles Labesse (CBS, Montpellier) for helping in mGlu2 modeling; Suren Felekyan (U. Dusseldorf, Germany) for discussion on data analysis; Guillaume Lebon, Alexandre Bouyssou (IGF, Montpellier, France) and the members of the IBM team (CBS, Montpellier) for fruitful discussions; the Arpège platform (IGF, Montpellier) for providing facilities and technical support, Perkin Elmer Cisbio for providing reagents. Our research is supported by grants from the Agence Nationale pour la Recherche (ANR 17-CE09-0026-02 to EM and ANR 18-CE11-0004-02 to EM and JPP), and the Fondation Recherche Médicale (DEQ20170336747 to JPP). AMC was supported by the Labex EPIGENMED. The CBS belongs to the France-BioImaging national infrastructure supported by the French National Research Agency (ANR-10-INBS-04, "Investments for the Future") and is supported by the GIS "IBiSA: Infrastructures en Biologie Santé et Agronomie".

## Author contributions

Anne-Marinette Cao: Detergent screening, protein preparation and purification, preliminary LRET and smFRET experiments, revised the manuscript. Robert B. Quast: Detergent optimization, protein preparation and purification, LRET and smFRET data acquisition and analysis, wrote the manuscript. Fataneh Fatemi: Preliminary protein preparation and purification and smFRET experiments, revised the paper. Philippe Rondard: Experiment design, data interpretation, wrote the paper. Jean-Philippe Pin: Experiment design, data interpretation, acquisition of funds, wrote the paper. Emmanuel Margeat: Experimental design, data analysis and interpretation, design, and implementation of the smFRET setup, acquisition of funds, wrote the paper.

## Competing interests

The authors declare no competing interests.
