## [Peer Review File · Nature Communications]

Allosteric modulators enhance agonist efficacy by increasing the residence time of a GPCR in the active stateEditorial Note: Parts of this Peer Review File have been redacted as indicated to remove third party material where no permission to publish were obtained

REVIEWER COMMENTS

Reviewer #1 (Remarks to the Author):

This is a very interesting manuscript that uses confocal based single molecule FRET measurements of mGlu2 in optimized detergent micelles to show that glutamate only partially stabilizes the extracellular domains in the active state and that addition of the PAM BINA or Gi is able to achieve full activation. The most salient breakthrough in this manuscript relates to the identification of solubilization conditions that maintain protein stability for the extended periods necessary for the types of confocal measurements that are made. Importantly, these conditions enable the authors to see PAM efficacy (BINA specifically), which is demonstrated to be modulated by the types of solubilization methods/components that are used. This is an important contribution to the literature.

While I found the manuscript overall to be quite compelling, there are a number of issues with the analyses and presentation that should be addressed:

1] While Fig.2 seems to make the overall point about the detergents, there are substantial differences between panel Fig. 2b (membrane fraction) and 2e,f,g in terms of the amplitudes of the LRET changes observed that should be clarified – what is the physical interpretation that explains why the amplitudes are lower in membranes than in detergent? Doesn't this imply something artificial about the detergent environment or something strange about the behaviors of the fluorophores employed in one or both settings?

2] The authors state that mGluR2 exhibits an intermediate FRET value in the apo state that is simply an average value of the fast exchange between high and low FRET states (~ 0.9 and ~ 0.3 respectively) (Fig.3). That the apo state is dynamic is supported by the analysis of the data presented. The state at 0.1 is fit – and apparently considered a distinct conformational state as it is first described, but it seems most likely that this state corresponds to zero FRET (likely due to photophysical blinking of the d2 fluorophore).

The authors should provide the structure of this fluorophore and information about the fluorescence parameters of the fluorophores that were used in the corrections that they state underpinning the FRET-corrections that they indicate were made.

In “smFRET data analysis” the authors state that “For some measurements, a minority of contaminating molecules with a long donor lifetime (>3 ns) were removed, as well as those with a lifetime >4 ns in the acceptor channel for measurements in IGEPAL”. How were these boundaries chosen? How different do the data look without such arbitrary selection? No quantitative information is given regarding how many molecules were excluded and what percentages of molecules remained in their final analyses. In this context, it seems relevant to examine how BINA and other exogenously added small-molecule ligands (whose structures should also be provided) affect fluorophore performances and/or the number of molecules passing whatever criteria were employed prior to data analyses. These details should be explicitly clarified somewhere.

Data for all histograms are recorded with a PIE excitation scheme, allowing selection of only molecules with an active donor and acceptor for the analysis. Representative 2D stoichiometry vs. FRET efficiency plot should be shown for every condition and with a scaling that highlights even low-abundance populations. Furthermore, the range of stoichiometry values used to select the data shown in Fig. 3b-e and 5e-g of the main text should be indicated explicitly. These additions would also help the reader to independently assess whether the small population at ≤ 0.1 FRET efficiency is a real population or a leftover from incomplete removal of the Donor-only population.

FRET efficiency histograms in Fig. 4d-f report a normalized occurrence to compare between different bin widths of the time window analysis. This makes the figure clearer but omits important information

about the number of events that are actually being used to construct the histograms, which is paramount to evaluating the robustness and significance of the resulting FRET efficiency distribution. Histograms reporting the actual number of molecules for every integration time should be shown in the same figure, or at least the number of molecules should be indicated in the figure or the caption and the histograms be put in the SI. More to this point, threshold number of photons used to define an event should be indicated for every integration time (and indeed every histogram recorded); if a threshold of 40 photons applies to every integration time value, this should be explicitly stated.

3] A major take home message of the work is that their measurements are physiologically relevant whereas previous measurements – primarily by the Isacoff lab – are not. The authors infer that because the Isacoff lab did not utilize the LMG-CHS-GDN solubilization – and because the Isacoff lab appears to report full activation in the presence of glutamate alone- that the system examined by Isacoff and colleagues must somehow be reporting on something that does not recapitulate important aspects of the physiology – artifacts of the solubilization conditions in particular. Yet, in the absence of very explicit clarifications, it seems theoretically possible that the constructs employed and/or the conditions of the experiments in the two labs give rise to the apparent disparities. In this regard, no specific mention of the conditions of the experiments can be found in the manuscript (oxygen scavenging, triplet state quenching additives, etc.). The mGluR2 construct is stated to come from Cisbio - what is the exact sequence, species, SNAP tag linker, and how does this compare to that used by the Isacoff lab. Without these details and careful comparisons it is difficult to rule out their potential contribution to the observed differences. Hence, it seems prudent to either make these comparisons explicitly before inferring that prior measurements were non-physiological or to substantially tone down the language in this aspect of the conclusions. It would also be interested to compare the results to the recent smFRET observations of mGlu2 in living cells in the presence and absence of glutamate (Asher et al, Nature Methods, 2021) as no detergent was present in these studies.

4] Where did GDN come from – was it just based on one example of it doing something beneficial previously? What does it do specifically? And why are the authors so convinced that it acts to preserve biological activity rather than simply perturb biological activity in a manner that allows them to see the influence of PAMs? Some additional discussion and whether GDN was resolved in the mglur5 structure should be added to the discussion. Also whether anything about the structural role of CHS in mGluR structures is worth discussing Additional discussion of the reported fluorescent contamination of these compounds is worth adding to the SI as it might help inform and guide other investigators who seek to replicate and build on this work.

5] I think the authors go too far in their overarching claim that PAMs simply modulate pre-existing GPCR dynamics based on distribution of FRET states. While this seems to be the most parsimonious model, how can the authors be certain that PAMs do not stabilize receptors in new conformations rather than modulate the rates and stabilities of pre-existing native states- the width of the FRET distributions appear broad enough to leave open the possibility that PAMs do in fact modify the structures in some manner. I would say the data may be consistent with the latter model, but I would not say the experiments definitively resolve the matter.

In this context, the authors also infer that the effect comes from modulating the dimer interface. This is possible but seems no more likely than impacting the binding sites themselves and their propensity of the extracellular domains to adapt conformations when binding glutamate. This discussion should also be tempered/broadened until more experimental data are available to resolve these fascinating questions.

Reviewer #2 (Remarks to the Author):

In this manuscript, Cao and colleagues have provided an experimental analysis of the sub-millisecond conformational dynamics of the ligand binding domain (LBD) of metabotropic glutamate receptor 2 in the context of the full length mGlu2 dimeric receptor. As noted by the authors, the ability of allosteric modulators to affect receptor activation/stabilization within this family of receptors is of critical importance to understanding their physiologic roles and in drug development. Other studies have begun to address mGlu receptor dynamics using single molecule LRET technology, and several have used full length receptor constructs; however, this study is the first to determine the influence of allosteric modulators and G protein binding on LBD dynamics. The authors have also provided a thorough and important methodological description of their receptor solubilization, which may influence future studies into solubilized mGlu receptors. Importantly, the solubilization and LRET analyses are unique to this study over other single molecule studies, allowing for important comparisons between isolation methods and validation of previously published findings. Lastly, this work provides important information into the mechanism of partial agonism and allostery provided by mGlu receptor modulators binding to the transmembrane domain. This reviewer recommends this manuscript be accepted with major revisions, particularly to the Supplemental Figures section, as well as statistical measures, as outlined below.

Specific points:

A. In the introduction, it is stated "...we optimized the conditions to conserve PAM activity at the solubilized full-length mGlu2 receptor and measure single molecule..." It should be noted that this is a chimeric (SNAP/CLIP tagged) receptor and should be stated as such.

B. It is noted within the manuscript that fluorescent contaminants were present within GDN and CHS preparations. Please state the absorbance and emission maxima for these preparations. It should be stated whether these fall within or outside the measurement settings of the smFRET analysis. If overlapping, can this fluorescence contribute within the kinetic window of the assay?

C. Perhaps the most interesting/confusing aspect of this manuscript is the invariable very high and very low FRET populations. As the authors stated, these populations could interconvert on a shortened time scale with those of the inactive (high FRET) and active (low FRET) pool of receptors. It seems possible that this could also represent a proportion of receptors which are not responsive in this assay, perhaps due to solubilization methods.

a. Can the stoichiometry factor distinguish between single, correctly labeled Donor-Acceptor dimeric pairs and oligomers, such as a dimer of dimers, where FRET between subunits would become more complicated? If SNAP and CLIP receptors are used, which would more absolutely select for strictly labeled dimers, are the results the same as what might be found in a situation in which dimers of dimers might occur?

b. Other smFRET studies have described 2-3 states: high FRET (inactive, open-open), low FRET (active, closed-closed), and a "middle FRET" state (inactive, open-closed) that is transient between the active and inactive state. How do the 4 states which are proposed relate to the open-open, open-closed, and closed-closed receptor states?

c. Within the literature, smFRET studies of mGlu receptors have been performed using integration times of 30-100 ms. This time is longer than the 5 ms time scale of the assay employed in this study. The very high fret and very low peaks are not present within these studies. The argument here is presumably that the increased specificity in integration time has captured a new state which is missed by other methods, much like the interconversion mentioned in the 0.2ms integration in Figure 4D. Have the authors performed an experiment with an increased integration time? Can the data be pooled after the experiment to more accurately compare to previously described high and low FRET states? This should be clearly discussed.

D. In Figure 4 a-c, is there a more quantitative way to represent the "skew", i.e., residuals?

E. Additional clarity is needed in the experiments involving antagonists. In Figure 3H, the addition of antagonist does not alter the receptor state distribution, contrary to Vafabakhsh 2015 where the low FRET state was abolished by LY34. These previously published data make sense if closure of the LBD is needed for reorientation of the LBDs relative to one another. This also agrees with data from Doumazane et al 2013 in which agonist was required for transition to the activated, low FRET LBD

state in full length receptors, and which was blocked entirely by addition of LY34. The data within the current manuscript suggest that LBD reorientation may still take place while the LBD open confirmation is stabilized by antagonist. This would suggest that agonist binding and LBD transition are effectively separate processes. It could be argued that these data, along with the presence of very high and very low FRET states, may represent an artifact specific to the experimental system and should be specifically addressed within the discussion.

F. How do these data relate to PAMs which bind the TMD but display agonism as well as allosteric activity? It is important for this technique to establish that differences observed in full length receptors by more traditional methods with unique allosteric modulators can be corroborated (or refuted) in this methodologic paradigm.

G. The Supplemental section is extremely confusing. For example, the Ro64 compound is mentioned in Supp Figure 9, but not discussed in the text in reference to this figure. Similarly, DCG-IV and Ro64 are also used in Supp Figure 12, but data with these compounds is also not discussed in the text. A discussion regarding Supplemental Figure 15 appears before Supplemental Figure 13; similarly, Figures S17-18 are discussed before Figure S16. While the Gi condition in Figures S17 and 18 is mentioned, the other conditions tested in Figures S17-18 are not discussed. There is also a Figure S19 discussed in the text that is not in the Supplemental Figure file. It is also suggested to move Figure 5c to the end of Figure 5 due to the flow of the results section regarding this figure. The order of many figures should be rearranged, and it is suggested that the authors should walk through each panel, in order, with each experimental condition discussed in the main body of the paper. With 19 supplemental figures, a clear presentation becomes even more imperative. The authors (end editors) should also consider whether the figures pertaining to the distinct detergent conditions could be consolidated into a single table, perhaps with two figures that show divergent effects of detergents left in as examples.

H. In the Figures, there are no statistical comparisons between any of the conditions. These need to be included; again, a table may be a better way to represent all of the data in the supplemental figures showing the various detergent conditions.

Minor point:

-There are a number of punctuation and grammatical errors throughout the manuscript that should be corrected.

Reviewer #3 (Remarks to the Author):

In this work, the authors conducted a study of the conformational mechanism of positive allosteric modulator (PAM) action on a metabotropic glutamate receptor (mGlu2) using single-molecule FRET (smFRET). The authors evaluate detergent conditions that best represent near native conditions by performing LRET measurements. With these measurements they identify that LMNG-CHS-GDN mixture is necessary to maintain the functional integrity of mGlu2. Under these conditions they found that the agonist glutamate, causes a partial shift from a high-FRET conformation to a low-FRET conformation, and that the presence of PAM or Gi protein in combination with glutamate causes a complete shift of the receptors to the low-FRET state. Similar measurements were also performed for investigating partial agonist activation of the receptor, and a similar partial shift of the FRET population was observed. These measurements are different from those previously reported for mGluR2 (Vafabakhsh et al, Nature 2015) and suggest that the importance of time resolution and detergent conditions.

Major concerns:

There is no discussion on what the four FRET states could represent, and no explanation is provided for combining the low FRET and very low FRET, as well as high FRET and very high FRET. This needs to be expanded more. No attempt is made to define structural changes that could be associated with these FRET states.

The authors show FRET histograms that clearly indicate a shift from a high-FRET population to a low-

FRET population when glutamate (and especially glutamate + BINA) is added. However, there is no discussion of the actual distances being measured. What are the inter-fluorophore distances that correlate to the four identified FRET states in the histograms? Are these distances physiologically possible? What were the expected distance changes based on previous structural studies of the mGlu2 dimer, and do the shifts seen here match the expected shifts in distance?

Only cumulative histograms are provided. Day to day histograms, histograms showing shifts due to agonist concentrations should be provided in supplemental for glutamate in absence and presence of PAMs.

In the final paragraph of section "Optimization of detergent conditions to obtain fully functional mGlu2 dimers", it is stated that the GDN and CHS used in this study contained contaminating traces of fluorescent substances, but no further explanation is given. The authors need to address this concern and evaluate the magnitude of the effect of these contaminants on the final results. Data collected from unlabeled mGlu2 molecules with GDN and CHS present in the solubilization could be used to determine the effect of fluorescent contaminants on the final data.

The large size of the snap tag used to conjugate the fluorophores to the mGlu2 protein needs to be addressed. Being almost 20kD, the snap tag is of a size that is non-negligible compared to the mGlu2 dimer. This creates a problem where the tag itself can alter the conformation of the protein of interest or can bias the location or orientation of the fluorophore. The possible effects of this large fluorophore attachment tag need to be discussed, and any possible effects should be accounted for in the final data analysis.

REVIEWER COMMENTS

Reviewer #1 (Remarks to the Author):

This is a very interesting manuscript that uses confocal based single molecule FRET measurements of mGlu2 in optimized detergent micelles to show that glutamate only partially stabilizes the extracellular domains in the active state and that addition of the PAM BINA or Gi is able to achieve full activation. The most salient breakthrough in this manuscript relates to the identification of solubilization conditions that maintain protein stability for the extended periods necessary for the types of confocal measurements that are made. Importantly, these conditions enable the authors to see PAM efficacy (BINA specifically), which is demonstrated to be modulated by the types of solubilization methods/components that are used. This is an important contribution to the literature.

>> We thank the reviewer for its careful reading of our manuscript, and recognizing the importance of our work.

While I found the manuscript overall to be quite compelling, there are a number of issues with the analyses and presentation that should be addressed:

1] While Fig.2 seems to make the overall point about the detergents, there are substantial differences between panel Fig. 2b (membrane fraction) and 2e,f,g in terms of the amplitudes of the LRET changes observed that should be clarified – what is the physical interpretation that explains why the amplitudes are lower in membranes than in detergent? Doesn't this imply something artificial about the detergent environment or something strange about the behaviors of the fluorophores employed in one or both settings?

>> Indeed, the amplitude of the LRET change is lower and the error in EC₅₀ is larger in membrane fractions as compared to receptors in detergent preparations. We believe that this effect stems from the presence of residual glutamate in the membrane fraction that is eliminated after receptor solubilization in detergent. Such an effect would be sufficient to explain the lower LRET efficiency (as the agonist decreases LRET efficacy), associated with the large variability observed in membranes with no glutamate added (as the amount of residual glutamate is not controllable and likely variable between membrane preparations). Consistently, we observed that the competitive antagonist LY34 leads to a slight increase in LRET in the absence of added glutamate in membrane fraction (Fig. S12) but not on receptors solubilized in LMNG-CHS-GDN (Fig. S13). Note that the presence of residual glutamate has been observed as well before by LRET measurement in living cells (Doumazane et al. 2013)

>> We note nevertheless that an effect of conditions on dyes photophysics cannot not be excluded. But although this would change the absolute LRET value, it would not influence the EC₅₀ that serves as our readout to compare the various detergent conditions with the membrane preparations.

2] The authors state that mGluR2 exhibits an intermediate FRET value in the apo state that is simply an average value of the fast exchange between high and low FRET states (~0.9 and~ 0.3 respectively) (Fig.3). That the apo state is dynamic is supported by the analysis of the data presented. The state at 0.1 is fit – and apparently considered a distinct conformational state as it is

first described, but it seems most likely that this state corresponds to zero FRET (likely due to photophysical blinking of the d2 fluorophore).

>> We think that this state probably represents a real conformational state as it appears with a Stoichiometry (S) value around 0.5, which indicates that both donor and acceptor are emitting photons. Indeed, if this state was due to an “inactive” (blinking) acceptor it would appear with a higher S value, the extreme limit being $S = 1$ if the acceptor is totally “off” during the transition of the molecule in the observation volume. As suggested by the reviewer below, we think the addition of 2D E - S histograms now allows the reader to judge that this population does not stem from a leftover of incomplete removal of the donor-only population.

The authors should provide the structure of this fluorophore and information about the fluorescence parameters of the fluorophores that were used in the corrections that they state underpinning the FRET-corrections that they indicate were made.

>> The structure of d2 is protected by Cis-Bio, the company selling the SNAP-labeling kits (<https://fr.cisbio.eu/d2-labeling-kit-40378>). Nevertheless, we know it is a Cy5 derivative, and have now indicated this in the main text.

“We then turned to the single molecule study of full-length mGlu2 and therefore substituted the LRET fluorophores with SNAP-tag substrates of Cy3B as donor and d2 (a Cy5 derivative) as acceptor.”

>> Concerning the values for the corrections factors used to calculate the FRET efficiencies, they are now indicated in the Material and Methods Section

“The values used for these corrections were $\alpha = 0.75$, $\delta = 0.02$ ”

In “smFRET data analysis” the authors state that “For some measurements, a minority of contaminating molecules with a long donor lifetime (>3 ns) were removed, as well as those with a lifetime >4ns in the acceptor channel for measurements in IGEPAL”. How were these boundaries chosen? How different do the data look without such arbitrary selection?

No quantitative information is given regarding how many molecules were excluded and what percentages of molecules remained in their final analyses. In this context, it seems relevant to examine how BINA and other exogenously added small-molecule ligands (whose structures should also be provided) affect fluorophore performances and/or the number of molecules passing whatever criteria were employed prior to data analyses. These details should be explicitly clarified somewhere.

>> We apologize for not making this point clear. The molecules with a long lifetime (>3ns) were only removed for “donor-only” species at the step where we determined accurately the “Donor Leakage” correction factor α (fraction of photons emitted by the donor detected in the acceptor channel). This fraction of molecules with $t > 3$ ns represented a few % of the total D-only molecules and this selection was made only for some condition. But importantly, after this α determination was made, this selection criterium was not applied anymore and all molecules were considered for further data display and analysis.

>> Concerning the measurements made in IGEPAL, these showed indeed a significant contamination by fluorescent species. We have now added a supplementary Figure (Fig. S26) to report on this observation and now clearly indicate in the material and methods the procedure used.

>> Additionally, we have rephrased this paragraph in the Material and Methods as follows.

“In LMNG-CHS-GDN minor contaminations of molecules appearing as donor-only, with a lifetime $\tau_D > 3\text{ns}$ were observed. These molecules were removed from our analysis solely to determine the donor leakage α factor but considered in all further analysis. In the presence of IGEPAL, significant contaminations of molecules with a lifetime $\tau_A > 2\text{ns}$ and molecules appearing as donor-only with a lifetime $\tau_D > 3\text{ns}$ (Fig. S26) were observed. Molecules with $\tau_D > 3\text{ns}$ were removed only for determination of α while molecules with $\tau_A > 2\text{ns}$ were completely removed from our analysis.”

>> We did not see any effect of the ligands or the detergents on the fluorophores properties. We have now stated this explicitly in the Material and Methods section

“Note that we did not see any effect of the ligands or the detergents on the fluorophores properties such as excited state lifetime (as measured for donor and acceptor) or relative brightness / quantum yield (as measured by determination of the γ factor)”

>> Concerning the structure of the molecules used, they are all publicly available on the website of Tocris Biosciences.

Data for all histograms are recorded with a PIE excitation scheme, allowing selection of only molecules with an active donor and acceptor for the analysis. Representative 2D stoichiometry vs. FRET efficiency plot should be shown for every condition and with a scaling that highlights even low-abundance populations.

>> The 2D histograms are now presented for all conditions in Supplementary Figure S14

>> Accordingly, we have added the following paragraph in the main text:

“A 2D histogram representation was used, where the X-axis represents the apparent FRET efficiency (E_{PR}) and the Y-axis the stoichiometry factor S calculated for each single molecule (Fig. S14) (Hellenkamp et al. 2018; Kapanidis et al. 2004). For further analysis only donor-acceptor (D-A) containing complexes were selected, based on the S factor ($0.3-0.35 < S < 0.6-0.65$).”

Furthermore, the range of stoichiometry values used to select the data shown in Fig. 3b-e and 5e-g of the main text should be indicated explicitly.

>> The range of S -values are now indicated in the Material and Methods

“To display the 1D FRET efficiency histograms and for further analysis, doubly labeled (Donor-Acceptor) molecules were selected using the stoichiometry S factor using ($0.3-0.35 < S < 0.6-0.65$).”

These additions would also help the reader to independently assess whether the small population at $E \leq 0.1$ FRET efficiency is a real population or a leftover from incomplete removal of the Donor-only population.

>> We agree with the reviewer that displaying the E - S histograms might help the reader judge whether this population at $E < 0.1$ comes from incomplete removal of the Donor-only population. We think that our E - S histograms (now presented in Fig. S14) show a very good separation between

species (we were particularly careful to work at very low receptor concentration to limit the simultaneous presence of 2 molecules in the observation volume). In any case, we would like to underline that whether or not this population represents a real conformational state does not influence the conclusions of our study on the effect of detergents and allosteric modulators on the conformational dynamics of full-length mGlu2.

FRET efficiency histograms in Fig. 4d-f report a normalized occurrence to compare between different bin widths of the time window analysis. This makes the figure clearer but omits important information about the number of events that are actually being used to construct the histograms, which is paramount to evaluating the robustness and significance of the resulting FRET efficiency distribution. Histograms reporting the actual number of molecules for every integration time should be shown in the same figure, or at least the number of molecules should be indicated in the figure or the caption and the histograms be put in the SI.

More to this point, threshold number of photons used to define an event should be indicated for every integration time (and indeed every histogram recorded); if a threshold of 40 photons applies to every integration time value, this should be explicitly stated.

>> Unfortunately, the exact number of events used to construct the histograms is not provided by the PAM program that we have been using for this analysis. We would like to stress that these histograms were constructed from a large number (typically > 5000) of molecules. We have chosen a relatively low photon threshold (25 photons) to make sure that the number of events for each histogram is significant. We would also like to stress out that we do not see differences in the photophysical behavior of the dyes for the various conditions. Thus, it is likely that the differences seen in the TWA analysis between the different ligands do not stem from a bias in the number of molecules taken into consideration for each time window.

>> We have now stated in the Material and Methods section the number of photons used for this analysis.

"A minimal threshold of 25 photons per time bins was used in the time windows analysis."

3] A major take home message of the work is that their measurements are physiologically relevant whereas previous measurements – primarily by the Isacoff lab – are not. The authors infer that because the Isacoff lab did not utilize the LMG-CHS-GDN solubilization – and because the Isacoff lab appears to report full activation in the presence of glutamate alone- that the system examined by Isacoff and colleagues must somehow be reporting on something that does not recapitulate important aspects of the physiology – artifacts of the solubilization conditions in particular. Yet, in the absence of very explicit clarifications, it seems theoretically possible that the constructs employed and/or the conditions of the experiments in the two labs give rise to the apparent disparities. In this regard, no specific mention of the conditions of the experiments can be found in the manuscript (oxygen scavenging, triplet state quenching additives, etc.). The mGluR2 construct is stated to come from Cisbio - what is the exact sequence, species, SNAP tag linker, and how does this compare to that used by the Isacoff lab.

>> We thank the reviewer for pointing out that our sequence was indeed not provided. It is now presented (Fig. S25) and a sentence in the Material and Methods section has been modified:

"The pcDNA plasmid encoding SNAP-tagged human mGlu2 was a gift from Cisbio Bioassays (Codolet, France) and is described in Fig. S25."

>> The construct used by the Isacoff lab was obtained from our lab, as stated in (Vafabakhsh, 2015). They used a very similar SNAP construct to the one used in our study but derived from the rat receptor, which shares 97.8% identity with the human receptor we used. Note that none of the Isacoff lab member publications report a clear characterization of an effect of a small molecule PAM or the G protein. Nevertheless, in order to verify that the lack of stability of the receptor we observed does not arise from the construct we used, we repeated LRET experiments using the rat construct (identical plasmid as used in Vafabakhsh, 2015) in the presence of Glu and Glu+PAM on membranes and on receptors solubilized in IGEPAL, DDM-CHS, and LMNG-CHS-GDN. These new results are now presented in Fig. S23 and confirm our results obtained on the human receptor constructs. Indeed, they show that in the case of the rat construct also used by the Isacoff lab, the PAM effect is not observed either when the receptor is solubilized in IGEPAL or DDM-CHS (identical conditions to the the recently published smFRET study on mouse mGlu in Liauw, 2021). These data are now introduced in the main text at two positions as follows:

“Strikingly similar results were obtained for the full-length rat mGlu2 (Fig. S23) previously used in smFRET studies (R Vafabakhsh, Levitz, and Isacoff 2015; Levitz et al. 2016).”

and

“A similar observation was made for another SNAP-sensor based on the rat receptor (Fig. S23) that was used in previous LRET and single molecule studies (Levitz et al. 2016; Doumazane et al. 2013; R Vafabakhsh, Levitz, and Isacoff 2015).”

>> In terms of experimental condition, we now explicitly refer in the smFRET section of the Material and Methods to the acquisition buffer that was used (previously it was only mentioned in the “detergent solubilization” section). Specifically, we do not use any oxygen scavenging or triplet state quenching additives that could indeed impair receptor function, as these types of additives are not mandatory to perform single molecule FRET measurements on diffusing molecules. We have therefore modified the following sentence in the M&M section

“Samples were measured in acquisition buffer (20 mM Tris-HCl pH7.4, 118 mM NaCl, 1.2 mM KH₂PO₄, 1.2 mM MgSO₄, 4.7 mM KCl, 1.8 mM CaCl₂) with detergent and ligand concentrations as indicated in the text and in the absence of any oxygen scavenging system or triplet state quenchers.”

Without these details and careful comparisons it is difficult to rule out their potential contribution to the observed differences. Hence, it seems prudent to either make these comparisons explicitly before inferring that prior measurements were non-physiological or to substantially tone down the language in this aspect of the conclusions.

>> We think we have been fair and balanced in treating the previous observations made in the literature on solubilized receptors. We just report here the fact that, in our hands, we observe that allosteric modulation and therefore full functional integrity of the receptor is lost when the proper detergent mixture is not used, including (our new data) with the construct used in those previous studies. We note that neither allosteric modulation by small molecules was observed in those previous studies, nor any modulation by G proteins was reported. Finally, we underline the fact that it is important to perform a careful optimization of solubilization conditions, especially when working with membrane proteins.

It would also be interested to compare the results to the recent smFRET observations of mGlu2 in living cells in the presence and absence of glutamate (Asher et al, Nature Methods, 2021) as no detergent was present in these studies.

>> The Asher et al. study published while our manuscript was under review is a very important contribution to the field of mGlu2 structural dynamics. Therefore, we mention this study in our discussion now:

“Interestingly, in a very recent study using a construct highly similar to ours, the structural dynamics of mGlu2 were monitored using smFRET at the surface of living cells, with a lower time resolution (40ms)(Asher et al. 2021). Transitions from a high FRET ($E = 0.44$) to a low FRET ($E = 0.3$) state were observed upon glutamate binding. Notably, some transitions to a very high FRET state ($E = 0.88$) and to a lower FRET state (not reported by the authors) could be observed in some traces, and we hypothesize that these rarely populated states could correspond to the VHF and VLF states observed in our study.”

>> We note that it would have been very interesting to explore the influence of a PAM in the structural dynamics of mGlu2 in live cells but unfortunately such an experiment was not reported in (Asher 2021).

4] Where did GDN come from – was it just based on one example of it doing something beneficial previously? What does it do specifically? And why are the authors so convinced that it acts to preserve biological activity rather than simply perturb biological activity in a manner that allows them to see the influence of PAMs?

>> In the recent years, GDN has been more and more used in the field of GPCR research, notably for structural studies (see below). We are convinced that it does not perturb the biological activity of mGlu but on the contrary helps preserving the natural one for the following reasons:

- Experiments in live cells (Doumazane, 2013) and on membrane fractions (Figure 2b) display a PAM effect similar to the one observed in the presence of LMNG-CHS-GDN (Figure 2g).
- In the absence of GDN in the LMNG-CHS mixture (Figure 2f), the PAM effect is observed as well, indicating that GDN is not solely responsible for this effect. Nevertheless, this effect is lost after a few hours at room temperature, while it remains unchanged when GDN is added to the mixture at an optimized ratio underlining the positive effect of GDN in maintaining a fully functional receptor.

Some additional discussion and whether GDN was resolved in the mglu5 structure should be added to the discussion. Also whether anything about the structural role of CHS in mGluR structures is worth discussing

GDN is being used by many groups as detergent to solubilize/maintain class C GPCRs in solution. The full-length mGlu5 cryo-EM structure in the agonist bound states were resolved in detergent micelles composed of GDN and CHS (0.02% and 0.002% respectively). However, neither GDN nor CHS molecules could be identified in the published cryo-EM structures (PDB ID 6N52 and 6N51, respectively). Although our data highlight the importance of both GDN and CHS to obtain a fully PAM regulated receptor, suggesting a role of CHS on the structural organization and conformational dynamics of mGlu2, more work is needed to document this. Although some mGlu (1 and 5) have been proposed to be associated with lipid rafts (enriched in cholesterol), the specific role of cholesterol on the function of mGlu has not been studied. At the structural level, cholesterol molecules were found at a TM1 dimer interface in the crystal structure of the isolated 7TM of mGlu1

(Wu et al., Science 2014). However, this dimer interface does not correspond to the ones observed in the various states of the full length mGlu. However, this still suggests a possible direct interaction between cholesterol and the 7TM bundle of mGlu.

Additional discussion of the reported fluorescent contamination of these compounds is worth adding to the SI as it might help inform and guide other investigators who seek to replicate and build on this work.

>> As stated below, we now report on the presence of fluorescent contaminations, which are only significant in the presence of IGEPAL, in the Material and Methods section and show corresponding data in Fig. S26.

>> For clarity, we have now rephrased our sentence in the main text as follows

“Note that the GDN and CHS concentrations we used remained sufficiently moderate to not create a fluorescence background that would have been detrimental to our smFRET studies. Indeed, we found that detergent solutions were slightly contaminated with fluorescent species of unknown origin (also found in batches from different suppliers).”

5] I think the authors go too far in their overarching claim that PAMs simply modulate pre-existing GPCR dynamics based on distribution of FRET states. While this seems to be the most parsimonious model, how can the authors be certain that PAMs do not stabilize receptors in new conformations rather than modulate the rates and stabilities of pre-existing native states- the width of the FRET distributions appear broad enough to leave open the possibility that PAMs do in fact modify the structures in some manner. I would say the data may be consistent with the latter model, but I would not say the experiments definitively resolve the matter.

>> We agree that our findings do not rule out that the PAM might modify the structure in a way that is not probed by our current FRET sensor and that therefore our claim should be more moderated. Therefore, we have modified the last paragraph as follows:

“Overall, by identifying conditions under which the solubilized mGlu2 receptor conserves its modulation by a PAM and the G protein, we have been able to show that BINA can increase the population of active receptors in the presence of Glu. Our data point to a model where the increased efficacy observed with this PAM would arise from its ability to stabilize the active state already populated in the presence of orthosteric agonists. However, the validation of such a model would require a deeper investigation of the conformations sampled by mGlu2 by measuring distances and distances changes between its various structural modules, using the incorporation and the labeling of unnatural amino acids for example. Such studies will pave the way for a deeper understanding of how the structural dynamics of mGlu receptors as well as other membrane receptors regulate their function and may open up new routes for the development of fine-tuned therapeutics.

In this context, the authors also infer that the effect comes from modulating the dimer interface. This is possible but seems no more likely than impacting the binding sites themselves and their propensity of the extracellular domains to adapt conformations when binding glutamate. This discussion should also be tempered/broadened until more experimental data are available to resolve these fascinating questions.

We agree with the referee, and, as indicated in our manuscript, this was a proposition and not a conclusion. We have therefore added the following sentence to the discussion :

"However, one cannot exclude that the 7TM bundles through their contact with the CRDs, also allosterically stabilize the CRD and VFT dimer in their active orientation."

Reviewer #2 (Remarks to the Author):

In this manuscript, Cao and colleagues have provided an experimental analysis of the sub-millisecond conformational dynamics of the ligand binding domain (LBD) of metabotropic glutamate receptor 2 in the context of the full length mGlu2 dimeric receptor. As noted by the authors, the ability of allosteric modulators to affect receptor activation/stabilization within this family of receptors is of critical importance to understanding their physiologic roles and in drug development. Other studies have begun to address mGlu receptor dynamics using single molecule LRET technology, and several have used full length receptor constructs; however, this study is the first to determine the influence of allosteric modulators and G protein binding on LBD dynamics. The authors have also provided a thorough and important methodological description of their receptor solubilization, which may influence future studies into solubilized mGlu receptors. Importantly, the solubilization and LRET analyses are unique to this study over other single molecule studies, allowing for important comparisons between isolation methods and validation of previously published findings. Lastly, this work provides important information into the mechanism of partial agonism and allostery provided by mGlu receptor modulators binding to the transmembrane domain.

>> We thank the reviewer for careful reading of our manuscript and recognizing the importance of our work

This reviewer recommends this manuscript be accepted with major revisions, particularly to the Supplemental Figures section, as well as statistical measures, as outlined below.

Specific points:

A. In the introduction, it is stated "...we optimized the conditions to conserve PAM activity at the solubilized full-length mGlu2 receptor and measure single molecule..." It should be noted that this is a chimeric (SNAP/CLIP tagged) receptor and should be stated as such.

>> Our receptor is labeled with a SNAP-tag (not CLIP) at its N-terminus (but subunits in the receptor dimer). The sequence is now given in Fig. S25. The introductory sentence has been changes to:

"Here, we optimized the conditions to conserve PAM activity of the solubilized full-length human mGlu2 receptor, N-terminally labeled through a SNAP-tag in each of the subunits, and measured single molecule Förster resonance energy transfer (smFRET) at nanoseconds time resolution."

B. It is noted within the manuscript that fluorescent contaminants were present within GDN and CHS preparations. Please state the absorbance and emission maxima for these preparations. It should be stated whether these fall within or outside the measurement settings of the smFRET analysis. If overlapping, can this fluorescence contribute within the kinetic window of the assay?

>> At the concentration we used this contamination is minimal (see our answer to Reviewer 1, #4, last paragraph). Nevertheless, it might lead to the appearance of fluorescence bursts at apparent FRET efficiency close to zero and we cannot exclude that the population we observe at very low FRET efficiency partly arises from these impurities. Nevertheless, we do not draw any conclusion on the nature of this population and its presence does not change the conclusion of our study.

C. Perhaps the most interesting/confusing aspect of this manuscript is the invariable very high and very low FRET populations. As the authors stated, these populations could interconvert on a shortened time scale with those of the inactive (high FRET) and active (low FRET) pool of receptors. It seems possible that this could also represent a proportion of receptors which are not responsive in this assay, perhaps due to solubilization methods.

>> The presence of these populations is indeed puzzling and many hypotheses could be brought forward to explain their presence. We note however that in our 2014 study on the isolated mGlu2 VFTs (performed without detergent), ligand-invariant HF and LF states were observed as well. One explanation could be that they arise from alternative configurations of the SNAP-tag relative to the receptor but in our 2014 study, an experiment where the SNAP tag was replaced with the smaller ACP tag led to similar FRET distributions (Olofsson, 2014, Supp. Material).

>> We think the best evidence that these populations do not arise from solubilization artifacts comes from a very recent single molecule FRET study on SNAP-mGlu2 in live cells (using a construct highly similar to ours in Asher et al., 2021). Indeed, this study states that “*Rare transitions were also observed to a FRET state of ~0.84 (Fig. 2h and Extended Data) consistent with previous reports of a minor population of high-FRET state conformations in isolated mGluR2 LBD dimers (Olofsson, 2014).*”) As we do not find notable changes in these VHF and VLF population in our study, we assume that transitions to/from these states are slower than the time resolution of our measurements (<5ms).

[REDACTED]	Figure 2h from Asher et al., 2021. Note the transition to a very high FRET state (as stated by the authors) and to a very low FRET state as well.
-------------------	--

a. Can the stoichiometry factor distinguish between single, correctly labeled Donor-Acceptor dimeric pairs and oligomers, such as a dimer of dimers, where FRET between subunits would become more complicated? If SNAP and CLIP receptors are used, which would more absolutely select for strictly labeled dimers, are the results the same as what might be found in a situation in which dimers of dimers might occur?

>> As requested by Reviewer 1, we now present the 2D histograms for all conditions in Fig. S14. Dimers of dimers with a ratio of Donor and Acceptor fluorophores unequal to 1 would have an S value intermediate between Donor-only ($S = 1$), Acceptor-only ($S = 0$) and D-A complexes ($S = 0.5$), which would therefore be centered around 0.25 for D-A-A, 0.5 for D-D-A-A, 0.75 for D-D-A). As a result of our stochastic labeling approach using two SNAP-tags (one on each protomer of the mGlu2

dimer) we should statistically detect them if they were present at elevated levels. We note that no significant populations of such species are seen around 0.25 and 0.75 and in any case such populations would be largely eliminated from our analysis, where we select only molecules with $0.3 < S < 0.6$.

>> Finally, D-D-A-A complexes at $S = 0.5$ would have a brightness twice the one of D-A complexes. We therefore selected this population around $S = 0.5$ and measured the average count rate per molecule of the acceptor upon acceptor excitation (insensitive to FRET) as a function of FRET efficiency. If any of these populations was composed of D-D-A-A complexes, this count rate should be doubled. We did not observe any significant change in count rate. As an example, here are the values obtained for a representative data set obtained in the presence of glutamate.

Population	E range	Acceptor – Acceptor count rate (kHz)
VLF	0-0.2	9.5
LF	0.2-0.4	10.2
HF	0.4-0.7	10.3
VHF	0.8-1	10.2

>> Finally, we would like to stress that at the concentration we used (~10pM receptor dimer), the presence of dimers of dimers is very unlikely.

b. Other smFRET studies have described 2-3 states: high FRET (inactive, open-open), low FRET (active, closed-closed), and a “middle FRET” state (inactive, open-closed) that is transient between the active and inactive state. How do the 4 states which are proposed relate to the open-open, open-closed, and closed-closed receptor states?

>> Other studies have indeed identified a “middle FRET” state, corresponding to an “open-closed” configuration of the receptor’s VFTs. Specifically, in (Vafabakhsh, 2015) this state was populated at intermediate, subsaturating concentrations or was seen on single molecule time traces during the transition between the HF and LF state. In our case, this state was not specifically populated, as our experiments were performed at saturating concentration of ligands. Moreover, as with our technique we do not have access to smFRET time traces, we cannot specifically focus our attention on the transition step between the HF and LF state. In addition, the N-terminal FRET sensor used in previous as well as our study does not allow for a definite identification of the closed or open states of the VFTs but rather reports on the VFT intersubunit rearrangement. It will thus require additional FRET sensors to monitor the relationship of VFT closure and VFT intersubunit rearrangement.

>> Therefore, we think that the LF and HF states observed in our study correspond to the active (closed-closed) and resting (open-open) states respectively, and this is consistent with the ensemble studies performed with mutated mGlu2 receptors stabilized in either the R or the A VFT dimer states (Doumazane et al., PNAS 2013, Fig 4). The additional VHF and VLF states we observed in our study, where all molecules are considered and displayed, might have been present in other studies, but could have been overlooked as they represent minor populations that are not influenced by ligand and probably not interconverting with the other states at fast timescales. Moreover, the VLF state can wrongly be assigned to a donor-only state, as alternating laser excitation was not used in these other studies.

>> Further investigation, using for example alternative labeling strategies using smaller tags such as UAAs and other labeling positions will be required to identify the nature of these additional states.

c. Within the literature, smFRET studies of mGlu receptors have been performed using integration times of 30-100 ms. This time is longer than the 5 ms time scale of the assay employed in this study. The very high fret and very low peaks are not present within these studies. The argument here is presumably that the increased specificity in integration time has captured a new state which is missed by other methods, much like the interconversion mentioned in the 0.2ms integration in Figure 4D. Have the authors performed an experiment with an increased integration time?

>> As stated above, it is possible that the VLF and VHF were present in previous studies, but have been overlooked. One argument for this hypothesis is that VLF and VHF states have been transiently observed using single molecule FRET in living cells by the Blanchard/Javich group recently (Asher 2021).

>>In our case, it is unfortunately not possible to observe mGlu conformational dynamics for longer time periods. Our observation time is limited by the diffusion time of each single receptor through the confocal volume. An experiment with longer integration time would require mGlu2 immobilization without disturbing its structural dynamics together with confocal scanning for high temporal resolution measurement. This type of experiment would require a lot of optimization, control and instrumental development and goes beyond the scope of the present work.

Can the data be pooled after the experiment to more accurately compare to previously described high and low FRET states? This should be clearly discussed.

>> It is not clear to us what the reviewer means by “pooling” the data. For each condition, our histograms and dynamic analysis already come from the pooling of thousands of molecules but the observation time for each molecule remains limited to around 5 ms.

D. In Figure 4 a-c, is there a more quantitative way to represent the “skew”, i.e., residuals?

>> The representation of Figure 4 is the one classically used in the literature to represent this type of data. Nevertheless, we agree with the reviewer that a representation that highlights the deviation of the data from the static FRET line might more efficiently underline the differences observed between the ligands. We have therefore generated such a representation, now presented in Fig. S20, which nicely supports our conclusions obtained from the visual observation of Figure 4 a-c.

E. Additional clarity is needed in the experiments involving antagonists. In Figure 3H, the addition of antagonist does not alter the receptor state distribution, contrary to Vafabakhsh 2015 where the low FRET state was abolished by LY34. These previously published data make sense if closure of the LBD is needed for reorientation of the LBDs relative to one another. This also agrees with data from Doumazane et al 2013 in which agonist was required for transition to the activated, low FRET LBD state in full length receptors, and which was blocked entirely by addition of LY34. The data within the current manuscript suggest that LBD reorientation may still take place while the LBD open conformation is stabilized by antagonist. This would suggest that agonist binding and LBD transition are effectively separate processes. It could be argued that these data, along with the presence of very high and very low FRET states, may represent an artifact specific to the

experimental system and should be specifically addressed within the discussion.

>> The reviewer is right in pointing out that we do not see any effect of the antagonist LY34 on the Apo receptor. But we would like to point out that in the study of Vafabackhsh 2015, such an effect is not seen either in the case of mGlu2. It is only observed in Vafabackhsh 2015 when mGlu3 is monitored, a subtype that is reported to have a higher basal activity than mGlu2, and a much higher glutamate affinity (Tora et al. 2018), making it more sensitive to tens of nanomolar glutamate contamination in the media. In the case of Doumazane 2013, with LRET experiments performed on mGlu2 in cells, the observed antagonist activity probably comes from the presence of residual glutamate in the buffer that is displaced by LY34.

>> We note that in our experiments on membrane fractions, where some residual glutamate might be present, a slight effect of LY34 is observed (Fig. S12). Nevertheless, to demonstrate the effect of LY34 in our single molecule conditions, we have now introduced a new supplementary Figure, where we show that LY34 at 1mM displaces glutamate used at 100 μ M and therefore reverses its effect (Fig. S17).

>> Accordingly, we have modified the main text as follows.

“Finally, we noted that application of saturating concentrations of the competitive orthosteric antagonist LY341495 (LY34) led to a similar distribution as seen for the apo receptor (Fig. 3h). This antagonist is able to bind and properly reverse the effect of a subsaturating concentration of glutamate (100 μ M) (Fig. S17). This indicates that no basal receptor activity or residual Glu was observed in our preparations, which was further verified by titration with LY34 in LRET measurements (Fig. S13a-e).”

F. How do these data relate to PAMs which bind the TMD but display agonism as well as allosteric activity? It is important for this technique to establish that differences observed in full length receptors by more traditional methods with unique allosteric modulators can be corroborated (or refuted) in this methodologic paradigm.

>> It would indeed be interesting to explore the structural dynamics of mGlu in response to other molecules that bind to the 7TM domain but exert by themselves an agonist effect in the absence of orthosteric agonist (molecules so called Ago-PAM), Unfortunately, although these types of Ago-PAM are available for other mGlu subtypes such as mGlu4 (Rovira et al., FASEB J 2015) or mGlu5 (Noetzel et al., Mol Pharmacol 2012), such type of compounds have not been described and validated to our knowledge in the case of mGlu2. Such an investigation on other mGlu subtypes will be the topic of further studies.

G. The Supplemental section is extremely confusing. For example, the Ro64 compound is mentioned in Supp Figure 9, but not discussed in the text in reference to this figure. Similarly, DCG-IV and Ro64 are also used in Supp Figure 12, but data with these compounds is also not discussed in the text. A discussion regarding Supplemental Figure 15 appears before Supplemental Figure 13; similarly, Figures S17-18 are discussed before Figure S16. While the Gi condition in Figures S17 and 18 is mentioned, the other conditions tested in Figures S17-18 are not discussed. There is also a Figure S19 discussed in the test that is not in the Supplemental Figure file. It is also suggested to move

Figure 5c to the end of Figure 5 due to the flow of the results section regarding this figure. The order of many figures should be rearranged, and it is suggested that the authors the authors should walk through each panel, in order, with each experimental condition discussed in the main body of the paper. With 19 supplemental figures, a clear presentation becomes even more imperative. The authors (end editors) should also consider whether the figures pertaining to the distinct detergent conditions could be consolidated into a single table, perhaps with two figures that show divergent effects of detergents left in as examples.

>> We thank the reviewer for pointing out these discrepancies and errors and apologize for the confusion it might have generated. As we have added new Supplementary Figures, this section and the corresponding references in the text, have been completely renumbered, and we have carefully checked that the order and the references to the Figures are consistent. Concerning the presentation of the data for the detergents as curves instead of as a table, we would like to insist on providing the raw titration curves, which show the quality and the reproducibility of our data.

>> We have added descriptions as follows:

“Under these detergent conditions, a small but significant effect of the negative allosteric modulator (NAM) Ro64-5229 that reduces the pEC50 of glutamate in membranes (Fig. S12a-c) and live cells³⁰, was observed as well (Fig. S9a-c). In addition, we confirmed the effect of the partial agonist DCG-IV, previously shown to promote changes in the VFT intersubunit orientation to a lower extent than full agonists³⁰. Such partial effect, strongly potentiated by the addition of BINA, was indeed observed on the solubilized receptor (Fig. S13). Notably, the effects of DCG-IV in the absence or presence of NAM and PAM reflected those observed in crude membranes (Fig. S12d-f), thus confirming native-like ligand responsiveness of the detergent-solubilized receptor.”

“This effect was reversible, as addition of an excess of the NAM Ro64 to receptors after activation by 500 nM BINA + Glu decreased the fraction of active receptor to a similar level observed when only Glu + NAM were applied (Fig. 5i-j and S16b), being slightly below that observed in the presence of Glu alone (Fig. 3i).”

“Application of the NAM Ro64 in the presence of Glu did not alter these partially remaining dynamic behavior (Fig. S21c and 22c).”

>> We have further adapted various figures with regard to coherence and readability by increasing font size, displaying the same type of cartoons and logical order of the panels. Further we have made many corrections to the main text and figure legends with regard to grammatical errors, coherence for abbreviations used and renumbering of figures. All figures displayed should now be mentioned in the text. All changes made are highlighted in yellow.

H. In the Figures, there are no statistical comparisons between any of the conditions. These need to be included; again, a table may be a better way to represent all of the data in the supplemental figures showing the various detergent conditions.

>> We think that there are a few Figures where a statistical analysis might be required to underline the effect (or the lack of effect) of different ligands or experimental conditions, namely Figure 5i, S9, S12, and S13. Therefore, for some conditions reported on these Figures, where only a small difference between data was observed, we performed and now report a one-way ANOVA with Sidak

or Turkey's multiple comparisons test as indicated. The results of these tests are consistent with our interpretation of the data as reported in the main text.

Minor point:

-There are a number of punctuation and grammatical errors throughout the manuscript that should be corrected.

>> We thank the reviewer for reporting this and have tried to correct these errors as much as possible.

Reviewer #3 (Remarks to the Author):

In this work, the authors conducted a study of the conformational mechanism of positive allosteric modulator (PAM) action on a metabotropic glutamate receptor (mGlu2) using single-molecule FRET (smFRET). The authors evaluate detergent conditions that best represent near native conditions by performing LRET measurements. With these measurements they identify that LMNG-CHS-GDN mixture is necessary to maintain the functional integrity of mGlu2. Under these conditions they found that the agonist glutamate, causes a partial shift from a high-FRET conformation to a low-FRET conformation, and that the presence of PAM or Gi protein in combination with glutamate causes a complete shift of the receptors to the low-FRET state. Similar measurements were also performed for investigating partial agonist activation of the receptor, and a similar partial shift of the FRET population was observed. These measurements are different from those previously reported for mGluR2 (Vafabakhsh et al, Nature 2015) and suggest that the importance of time resolution and detergent conditions.

>> We thank the reviewer for careful reading of our manuscript, and recognizing the importance of our work

Major concerns:

The is no discussion on what the four FRET states could represent, and no explanation is provided for combining the low FRET and very low FRET, as well as high FRET and very high FRET. This needs to be expanded more. No attempt is made to define structural changes that could be associated with these FRET states.

>> We have indeed not tried to make a direct relationship between the observed FRET states and structural states derived from high resolution structural studies of mGlu receptors. Indeed, the use of SNAP tags precludes our ability to make such a direct relationship, due to the uncertainty in the positioning of the SNAP tag relative to the receptor and the lack of rotational freedom of the fluorophore within the SNAP tag. Therefore, we only assume, as backed by structural studies and all single molecule FRET studies published so far, that the receptor in Resting state (Resting orientation both VFTs open (Roo)) is in a high FRET state, consistent with the ensemble study (Doumazane et al., PNAS 2013, Fig 4), while FRET is reduced in the “active state” (active orientation, with both VFTs in the closed state (Acc state)). We decided to pool the VHF and HF population as “active” and VLF and LF as “active” in our analysis for Fig. 3i-j, and 5i-j. Nevertheless, we would like to point out, as shown in Figure 5j, that a similar result would have been obtained by neglecting VLF and VHF population (that barely vary in the presence of the various ligands) and only calculating the “Fraction active state” using $(LF/LF+HF)$ instead of $(VLF+LF)/(VLF+LF+HF+VHF)$.

The authors show FRET histograms that clearly indicate a shift from a high-FRET population to a low-FRET population when glutamate (and especially glutamate + BINA) is added. However, there is no discussion of the actual distances being measured. What are the inter-fluorophore distances that correlate to the four identified FRET states in the histograms? Are these distances physiologically possible? What were the expected distance changes based on previous structural studies of the mGlu2 dimer, and do the shifts seen here match the expected shifts in distance?

>> As stated above, the use of SNAP-tags unfortunately precludes extracting exact distances to correlate with high resolution structures. Nevertheless, in our pioneer study on the isolated VFT dimers using the same SNAP-tag construct (Olofsson 2014), we calculated accessible volume (AV) simulations of the fluorophore positions (dye clouds) based on the crystal structures of mGlu1 in the

active or resting conformations. The observed distribution in theoretical distances was very wide but the variation in distances following the reorientation of the VFT dimer was compatible with FRET measurements as a significant shift in the predicted distributions toward lower FRET values was observed.

[REDACTED]

Olofsson et al. 2014.
Supplementary Figure 4

>> We note that a careful examination of the existence of all four states in the absence of SNAP-tags would require the incorporation of unnatural amino acids as a handle for receptor labeling. The use of such advanced labeling strategies goes beyond the current work but is currently under investigation in our lab and by others (Liauw 2021).

Only cumulative histograms are provided. Day to day histograms, histograms showing shifts due to agonist concentrations should be provided in supplemental for glutamate in absence and presence of PAMs.

>> The presented histograms are from a representative experiment from a single biological replicate. For each dataset in our manuscript, at least 3 biological replicates have been used (representing the repetition of 3 completely independent experiments (from the cell transfection step at different cell passages through labeling, membrane preparation and receptor solubilization). This is what is represented by the different data points in Fig. 2, Fig. 3i-j and Fig. 5i-j. We would like to stress that the representation of several biological replicates underlines the robustness of our data, as such practice is seldom explicitly used in the field of smFRET.

>> In order to better report on the reproducibility of our data, we now exemplarily introduce Fig. S15, which presents smFRET histograms for 3 biological replicates at 3 different ligand conditions (Apo, Glu, Glu+BINA)

>> Accordingly, we now refer to this Figure in the text:

“FRET histograms of SNAP-labeled, full-length mGlu2 in LMNG-CHS-GDN micelles showed a wide, multimodal distribution (**Fig. 3b-h**), indicating the co-existence of four main VFT states. These distributions were observed reproducibly by repeating experiments on independent biological replicates of solubilized receptors originating from membrane fractions prepared at different cell passages (**Fig. S15**).”

In the final paragraph of section “Optimization of detergent conditions to obtain fully functional mGlu2 dimers”, it is stated that the GDN and CHS used in this study contained contaminating traces of fluorescent substances, but no further explanation is given. The authors need to address this concern and evaluate the magnitude of the effect of these contaminants on the final results. Data

collected from unlabeled mGlu2 molecules with GDN and CHS present in the solubilization could be used to determine the effect of fluorescent contaminants on the final data.

>> The effect of the contaminants on the final data is negligible. We have now clarified this point in our description of the data analysis procedure and have addressed it in detail in our answers to Reviewer 1.

The large size of the snap tag used to conjugate the fluorophores to the mGlu2 protein needs to be addressed. Being almost 20kD, the snap tag is of a size that is non-negligible compared to the mGlu2 dimer. This creates a problem where the tag itself can alter the conformation of the protein of interest or can bias the location or orientation of the fluorophore. The possible effects of this large fluorophore attachment tag need to be discussed, and any possible effects should be accounted for in the final data analysis.

>> The reviewer's comment is correct: the large size of the SNAP-tag and its possible reorientation might influence the measured FRET efficiencies. This is indeed why in our study we neither report any distance measurements, nor try to attribute the various observed states to a specific conformation of the protein obtained from high resolution structure. We only attribute the populations observed at high FRET to the "resting" state, and those observed at low FRET to the active state, as established by us and others on similar SNAP-tagged constructs (Doumazane 2013, Olofsson 2014, Vafabakhsh, 2015, ..., Asher 2021).

>> In addition, we would like to remind that it has been established that the SNAP-tagged receptors exert a pharmacological response identical to the WT receptor, as monitored in pharmacological assays (Doumazane 2013; Schoeller et al. 2017). Moreover, in our first smFRET study (Olofsson 2014) on the isolated mGlu2 VFTs, the various FRET populations and fast (sub-ms) structural dynamics observed using the SNAP-tag construct, were observed as well when the SNAP-tag was replaced with the much smaller ACP-tag.

REVIEWER COMMENTS

Reviewer #2 (Remarks to the Author):

We thank the authors for their substantial revision; however, a few more modifications are needed for clarity. Specific notes in regard to the submitted rebuttal answers follow a ">" symbol below.

A. In the introduction, it is stated "...we optimized the conditions to conserve PAM activity at the solubilized full-length mGlu2 receptor and measure single molecule..." It should be noted that this is a chimeric (SNAP/CLIP tagged) receptor and should be stated as such.

>> Our receptor is labeled with a SNAP-tag (not CLIP) at its N-terminus (but subunits in the receptor dimer). The sequence is now given in Fig. S25. The introductory sentence has been changed to: "Here, we optimized the conditions to conserve PAM activity of the solubilized full-length human mGlu2 receptor, N-terminally labeled through a SNAP-tag in each of the subunits, and measured single molecule Förster resonance energy transfer (smFRET) at nanoseconds time resolution."

>Acceptable for publication

B. It is noted within the manuscript that fluorescent contaminants were present within GDN and CHS preparations. Please state the absorbance and emission maxima for these preparations. It should be stated whether these fall within or outside the measurement settings of the smFRET analysis. If overlapping, can this fluorescence contribute within the kinetic window of the assay?

>> At the concentration we used this contamination is minimal (see our answer to Reviewer 1, #4, last paragraph). Nevertheless, it might lead to the appearance of fluorescence bursts at apparent FRET efficiency close to zero and we cannot exclude that the population we observe at very low FRET efficiency partly arises from these impurities. Nevertheless, we do not draw any conclusion on the nature of this population and its presence does not change the conclusion of our study.

> No conclusions may be drawn from these states in isolation; however, at least in one analysis the very low FRET state is combined with the low FRET state which is then used to draw conclusions. "We assigned the VHF and HF populations to a conformational ensemble representing the resting/inactive VFT state and the VLF and LF populations to the active state (Fig. 3b-h). Then, to gain a quantitative view of mGlu2 receptor activation, we calculated the fraction of active molecules found in the LF+VLF states relative to all molecules." Please see a continuation of this response in part C.

C. Perhaps the most interesting/confusing aspect of this manuscript is the invariable very high and very low FRET populations. As the authors stated, these populations could interconvert on a shortened time scale with those of the inactive (high FRET) and active (low FRET) pool of receptors. It seems possible that this could also represent a proportion of receptors which are not responsive in this assay, perhaps due to solubilization methods.

>> The presence of these populations is indeed puzzling and many hypotheses could be brought forward to explain their presence. We note however that in our 2014 study on the isolated mGlu2 VFTs (performed without detergent), ligand-invariant HF and LF states were observed as well. One explanation could be that they arise from alternative configurations of the SNAP-tag relative to the receptor but in our 2014 study, an experiment where the SNAP tag was replaced with the smaller ACP tag led to similar FRET distributions (Olofsson, 2014, Supp. Material).

>> We think the best evidence that these populations do not arise from solubilization artifacts comes from a very recent single molecule FRET study on SNAP-mGlu2 in live cells (using a construct highly similar to ours in Asher et al., 2021). Indeed, this study states that "Rare transitions were also observed to a FRET state of ~ 0.84 (Fig. 2h and Extended Data) consistent with previous reports of a minor population of high-FRET state conformations in isolated mGluR2 LBD dimers (Olofsson, 2014).) As we do not find notable changes in these VHF and VLF population in our study, we assume that transitions to/from these states are slower than the time resolution of our measurements (<5ms).

>These data from Asher et al, which track the dynamics of receptor complexes across time, suggest that the high FRET state occurs in functional receptors. These data also show though that this high FRET state is witnessed with a greater propensity with 15 μ M glutamate conditions than apo conditions (extended data figure 5C), suggesting that high FRET could be influenced by the presence of agonist and that this FRET state may not be related to an inactivated receptor state allowing it to be combined with the high FRET state thought to represent the relaxed receptor. Not enough is determined in the present study to allow for these states to be combined for analysis. Additionally, more discussion is needed into how these potentially interconvertible states could relate to mGlu receptor dynamics. Furthermore, in the analyses from Asher et al., the high fret state appears to be observed on millisecond to submillisecond time windows. Does the percentage of receptors within this high FRET state within the current analysis agree with what could be observed stochastically from many single molecules occupying this state so transiently? Some discussion to this point should be added.

a. Can the stoichiometry factor distinguish between single, correctly labeled Donor-Acceptor dimeric pairs and oligomers, such as a dimer of dimers, where FRET between subunits would become more complicated? If SNAP and CLIP receptors are used, which would more absolutely select for strictly labeled dimers, are the results the same as what might be found in a situation in which dimers of dimers might occur?

>> As requested by Reviewer 1, we now present the 2D histograms for all conditions in Fig. S14. Dimers of dimers with a ratio of Donor and Acceptor fluorophores unequal to 1 would have an S value intermediate between Donor-only ($S = 1$), Acceptor -only ($S = 0$) and D-A complexes ($S = 0.5$), which would therefore be centered around 0.25 for D-A-A, 0.5 for D-D-A-A, 0.75 for D-D-A. As a result of our stochastic labeling approach using two SNAP-tags (one on each protomer of the mGlu2 dimer) we should statistically detect them if they were present at elevated levels. We note that no significant populations of such species are seen around 0.25 and 0.75 and in any case such populations would be largely eliminated from our analysis, where we select only molecules with $0.30.35 < S < 0.6-0.65$.

>> Finally, D-D-A-A complexes at $S = 0.5$ would have a brightness twice the one of D-A complexes. We therefore selected this population around $S = 0.5$ and measured the average count rate per molecule of the acceptor upon acceptor excitation (insensitive to FRET) as a function of FRET efficiency. If any of these populations was composed of D-D-A-A complexes, this count rate should be doubled. We did not observe any significant change in count rate. As an example, here are the values obtained for a representative data set obtained in the presence of glutamate.

Population E range Acceptor – Acceptor count rate (kHz)

VLF 0-0.2 9.5

LF 0.2-0.4 10.2

HF 0.4-0.7 10.3

VHF 0.8-1 10.2

>> Finally, we would like to stress that at the concentration we used (~ 10 pM receptor dimer), the presence of dimers of dimers is very unlikely.

>Do the count rates shown represent those determined from pooled averages of molecules for each determined FRET efficiency range? If so, could there be dimer of dimers still in the analysis? Is the filtering method for fluorescent molecule detection able to detect this increase in acceptor count and exclude D-D-A-A complexes? While it can be appreciated that, at such a low concentrations a dimer of dimers would be unlikely to form, they could be formed within membranes and would likely be solubilized as a D-D-A-A complex. mGlu receptors are known to maintain dimerization in non-denaturing conditions and it is not expected that dilution of receptor complexes would change this.

b. Other smFRET studies have described 2-3 states: high FRET (inactive, open-open), low FRET (active, closed-closed), and a "middle FRET" state (inactive, open-closed) that is transient between the active and inactive state. How do the 4 states which are proposed relate to the open-open, open-

closed, and closed-closed receptor states?

>> Other studies have indeed identified a "middle FRET" state, corresponding to an "open-closed" configuration of the receptor's VFTs. Specifically, in (Vafabakhsh, 2015) this state was populated at intermediate, subsaturating concentrations or was seen on single molecule time traces during the transition between the HF and LF state. In our case, this state was not specifically populated, as our experiments were performed at saturating concentration of ligands. Moreover, as with our technique we do not have access to smFRET time traces, we cannot specifically focus our attention on the transition step between the HF and LF state. In addition, the N-terminal FRET sensor used in previous as well as our study does not allow for a definite identification of the closed or open states of the VFTs but rather reports on the VFT intersubunit rearrangement. It will thus require additional FRET sensors to monitor the relationship of VFT closure and VFT intersubunit rearrangement.

>> Therefore, we think that the LF and HF states observed in our study correspond to the active (closed-closed) and resting (open-open) states respectively, and this is consistent with the ensemble studies performed with mutated mGlu2 receptors stabilized in either the R or the A VFT dimer states (Doumazane et al., PNAS 2013, Fig 4). The additional VHF and VLF states we observed in our study, where all molecules are considered and displayed, might have been present in other studies, but could have been overlooked as they represent minor populations that are not influenced by ligand and probably not interconverting with the other states at fast timescales. Moreover, the VLF state can wrongly be assigned to a donor-only state, as alternating laser excitation was not used in these other studies.

>> Further investigation, using for example alternative labeling strategies using smaller tags such as UAAs and other labeling positions will be required to identify the nature of these additional states.

> It is of the opinion of this reviewer that the relation of observed FRET states to Open-Open and Closed-Closed states should be stated clearly in the discussion in order to relate these findings more clearly to other mGlu literature.

c. Within the literature, smFRET studies of mGlu receptors have been performed using integration times of 30-100 ms. This time is longer than the 5 ms time scale of the assay employed in this study. The very high fret and very low peaks are not present within these studies. The argument here is presumably that the increased specificity in integration time has captured a new state which is missed by other methods, much like the interconversion mentioned in the 0.2ms integration in Figure 4D. Have the authors performed an experiment with an increased integration time?

>> As stated above, it is possible that the VLF and VHF were present in previous studies, but have been overlooked. One argument for this hypothesis is that VLF and VHF states have been transiently observed using single molecule FRET in living cells by the Blanchard/Javich group recently (Asher 2021).

>> In our case, it is unfortunately not possible to observe mGlu conformational dynamics for longer time periods. Our observation time is limited by the diffusion time of each single receptor through the confocal volume. An experiment with longer integration time would require mGlu2 immobilization without disturbing its structural dynamics together with confocal scanning for high temporal resolution measurement. This type of experiment would require a lot of optimization, control and instrumental development and goes beyond the scope of the present work.

> We agree that these additional experiments would be beyond the scope of this study, but please see response to part B, C.

Can the data be pooled after the experiment to more accurately compare to previously described high and low FRET states? This should be clearly discussed.

>> It is not clear to us what the reviewer means by "pooling" the data. For each condition, our histograms and dynamic analysis already come from the pooling of thousands of molecules but the observation time for each molecule remains limited to around 5 ms.

> Originally, the reviewer was under the impression that the integration time could be increased to capture a distribution that more closely resembled other studies, suggesting more comparability

between this and other studies. It is now apparent that this is dependent on the current experimental design, and so the authors' response is understood.

D. In Figure 4 a-c, is there a more quantitative way to represent the "skew", i.e., residuals?

>> The representation of Figure 4 is the one classically used in the literature to represent this type of data. Nevertheless, we agree with the reviewer that a representation that highlights the deviation of the data from the static FRET line might more efficiently underline the differences observed between the ligands. We have therefore generated such a representation, now presented in Fig. S20, which nicely supports our conclusions obtained from the visual observation of Figure 4 a-c.

>Acceptable for publication

E. Additional clarity is needed in the experiments involving antagonists. In Figure 3H, the addition of antagonist does not alter the receptor state distribution, contrary to Vafabakhsh 2015 where the low FRET state was abolished by LY34. These previously published data make sense if closure of the LBD is needed for reorientation of the LBDs relative to one another. This also agrees with data from Doumazane et al 2013 in which agonist was required for transition to the activated, low FRET LBD state in full length receptors, and which was blocked entirely by addition of LY34. The data within the current manuscript suggest that LBD reorientation may still take place while the LBD open conformation is stabilized by antagonist. This would suggest that agonist binding and LBD transition are effectively separate processes. It could be argued that these data, along with the presence of very high and very low FRET states, may represent an artifact specific to the experimental system and should be specifically addressed within the discussion.

>> The reviewer is right in pointing out that we do not see any effect of the antagonist LY34 on the Apo receptor. But we would like to point out that in the study of Vafabakhsh 2015, such an effect is not seen either in the case of mGlu2. It is only observed in Vafabakhsh 2015 when mGlu3 is monitored, a subtype that is reported to have a higher basal activity than mGlu2, and a much higher glutamate affinity (Tora et al. 2018), making it more sensitive to tens of nanomolar glutamate contamination in the media. In the case of Doumazane 2013, with LRET experiments performed on mGlu2 in cells, the observed antagonist activity probably comes from the presence of residual glutamate in the buffer that is displaced by LY34.

>> We note that in our experiments on membrane fractions, where some residual glutamate might be present, a slight effect of LY34 is observed (Fig. S12). Nevertheless, to demonstrate the effect of LY34 in our single molecule conditions, we have now introduced a new supplementary Figure, where we show that LY34 at 1mM displaces glutamate used at 100 μ M and therefore reverses its effect (Fig. S17).

>> Accordingly, we have modified the main text as follows.

"Finally, we noted that application of saturating concentrations of the competitive orthosteric antagonist LY341495 (LY34) led to a similar distribution as seen for the apo receptor (Fig. 3h). This antagonist is able to bind and properly reverse the effect of a subsaturating concentration of glutamate (100 μ M) (Fig. S17). This indicates that no basal receptor activity or residual Glu was observed in our preparations, which was further verified by titration with LY34 in LRET measurements (Fig. S13a-e)."

>We understand the author's argument about antagonist blocking the effect of subsaturating glutamate. We agree that this assay seems to have low basal activity. However, even without glutamate present, antagonist will bind to the LBD and stabilize the open conformation. One would expect, if LBD closure is inherently linked to subunit rearrangement and alteration in FRET efficiency that this would result in an alteration in the distribution of FRET states, specifically with a reduction in the very low and low FRET states. The absence of this effect suggests a rearrangement that is independent of LBD closure and should be discussed.

F. How do these data relate to PAMs which bind the TMD but display agonism as well as allosteric activity? It is important for this technique to establish that differences observed in full length receptors

by more traditional methods with unique allosteric modulators can be corroborated (or refuted) in this methodologic paradigm.

>> It would indeed be interesting to explore the structural dynamics of mGlu in response to other molecules that bind to the 7TM domain but exert by themselves an agonist effect in the absence of orthosteric agonist (molecules so called Ago-PAM), Unfortunately, although these types of Ago-PAM are available for other mGlu subtypes such as mGlu4 (Rovira et al., FASEB J 2015) or mGlu5 (Noetzel et al., Mol Pharmacol 2012), such type of compounds have not been described and validated to our knowledge in the case of mGlu2. Such an investigation on other mGlu subtypes will be the topic of further studies.

>Acceptable for publication

G. The Supplemental section is extremely confusing. For example, the Ro64 compound is mentioned in Supp Figure 9, but not discussed in the text in reference to this figure. Similarly, DCG-IV and Ro64 are also used in Supp Figure 12, but data with these compounds is also not discussed in the text. A discussion regarding Supplemental Figure 15 appears before Supplemental Figure 13; similarly, Figures S17-18 are discussed before Figure S16. While the Gi condition in Figures S17 and 18 is mentioned, the other conditions tested in Figures S17-18 are not discussed. There is also a Figure S19 discussed in the text that is not in the Supplemental Figure file. It is also suggested to move Figure 5c to the end of Figure 5 due to the flow of the results section regarding this figure. The order of many figures should be rearranged, and it is suggested that the authors should walk through each panel, in order, with each experimental condition discussed in the main body of the paper. With 19 supplemental figures, a clear presentation becomes even more imperative. The authors (end editors) should also consider whether the figures pertaining to the distinct detergent conditions could be consolidated into a single table, perhaps with two figures that show divergent effects of detergents left in as examples.

>> We thank the reviewer for pointing out these discrepancies and errors and apologize for the confusion it might have generated. As we have added new Supplementary Figures, this section and the corresponding references in the text, have been completely renumbered, and we have carefully checked that the order and the references to the Figures are consistent. Concerning the presentation of the data for the detergents as curves instead of as a table, we would like to insist on providing the raw titration curves, which show the quality and the reproducibility of our data.

>> We have added descriptions as follows:

"Under these detergent conditions, a small but significant effect of the negative allosteric modulator (NAM) Ro64-5229 that reduces the pEC50 of glutamate in membranes (Fig. S12a-c) and live cells 30, was observed as well (Fig. S9a-c). In addition, we confirmed the effect of the partial agonist DCG-IV, previously shown to promote changes in the VFT intersubunit orientation to a lower extent than full agonists 30. Such partial effect, strongly potentiated by the addition of BINA, was indeed observed on the solubilized receptor (Fig. S13). Notably, the effects of DCG-IV in the absence or presence of NAM and PAM reflected those observed in crude membranes (Fig. S12d-f), thus confirming native-like ligand responsiveness of the detergent-solubilized receptor."

"This effect was reversible, as addition of an excess of the NAM Ro64 to receptors after activation by 500 nM BINA + Glu decreased the fraction of active receptor to a similar level observed when only Glu + NAM were applied (Fig. S1j and S16b), being slightly below that observed in the presence of Glu alone (Fig. 3i)."

"Application of the NAM Ro64 in the presence of Glu did not alter these partially remaining dynamic behavior (Fig. S21c and 22c)."

>> We have further adapted various figures with regard to coherence and readability by increasing font size, displaying the same type of cartoons and logical order of the panels. Further we have made many corrections to the main text and figure legends with regard to grammatical errors, coherence for abbreviations used and renumbering of figures. All figures displayed should now be mentioned in the text. All changes made are highlighted in yellow.

>Thank you for making these corrections.

In the Figures, there are no statistical comparisons between any of the conditions. These need to be included; again, a table may be a better way to represent all of the data in the supplemental figures showing the various detergent conditions.

>> We think that there are a few Figures where a statistical analysis might be required to underline the effect (or the lack of effect) of different ligands or experimental conditions, namely Figure 5i, S9, S12, and S13. Therefore, for some conditions reported on these Figures, where only a small difference between data was observed, we performed and now report a one-way ANOVA with Sidak or Turkey's multiple comparisons test as indicated. The results of these tests are consistent with our interpretation of the data as reported in the main text.

>There are still figures where statistics are not clearly marked. Statistical tests for each figure should be clearly denoted within the figure legend.

Ex. In figure 2 there is no mention of any statistical tests to denote differences in membrane preparations.

Ex. S23. It is noted that 95% confidence intervals are plotted but there is no information on what statistics were run or if any of the observed differences are significant.

Reviewer #3 (Remarks to the Author):

The authors have addressed my concerns.

REVIEWER COMMENTS

We thank the reviewers for their constructive input, and their recognition of our efforts to take into account their comments on the first version of the manuscript. Please find below our answer to their second round of suggestions and questions. Reviewers (and editor for reviewer 1) comments are in **bold italic**. Our answers are in plain text, and new modifications to the manuscript are highlighted in yellow.

Reviewer #1 (Remarks to the Editor):

Reviewer #1 had only limited availability to assess this revised version of the manuscript, so they don't provide a full-length report. However, in their remarks to the editors, they re-iterate the concerns regarding (1) the FRET efficiency histogram resulting from the binning at 0.2ms; (2) regarding Supplementary Figure 22 (where non-normalized plots showing number of events/photon in bins should be shown), and (3) regarding some of the answers to previous questions (in particular their previous point 1), which are presented in the point-by-point response document, but not added to the manuscript text.

It is very difficult for us to answer to the remarks of reviewer 1, without having access to a detailed report. We tried nevertheless to account for his concerns as follows :

(1) the FRET efficiency histogram resulting from the binning at 0.2ms; (2) regarding Supplementary Figure 22 (where non-normalized plots showing number of events/photon in bins should be shown)

We have now requested from the developers a modification of the PAM software, in order to be able to reports the number of data points used to build the histograms, notably at 0.2ms. This allows to account for a previous concern of the reviewer (***"This makes the figure clearer but omits important information about the number of events that are actually being used to construct the histograms, which is paramount to evaluating the robustness and significance of the resulting FRET efficiency distribution."***)

We now report in the Figure legend the number of events used to constructs the histograms. It shows that, in the worst case, we use at least ~900 events (at 0.2ms) to >12000 events (at 1ms), which are significant numbers and account for the robustness of our analysis.

(3) regarding some of the answers to previous questions (in particular their previous point 1), which are presented in the point-by-point response document, but not added to the manuscript text.

Looking at our point-by-point response to reviewer 1 in the first round of revision, it indeed appears that we did not include our answer to the point 1 in the manuscript. This concerned the ***"differences between panel Fig. 2b (membrane fraction) and 2e,f,g is terms of the amplitudes of the LRET changes observed that should be clarified – what is the physical interpretation that explains why the amplitudes are lower in membranes that in detergent?"***

A paragraph has now been added in the results section, as follows

"For all our data, we noted that the amplitude of the LRET change were lower and the error in EC50 larger in membrane fractions as compared to receptors in detergent"

preparations. We believe that this effect stems from the presence of residual glutamate in the membrane fractions that is eliminated after receptor solubilization in detergent and the important dilution. Such an effect would be sufficient to explain the lower LRET efficiency (as the agonist decreases LRET efficacy), associated with the large variability observed in membranes with no glutamate added (as the amount of residual glutamate is not controllable and likely variable between membrane preparations).

Reviewer #2 (Remarks to the Author):

We thank the authors for their substantial revision; however, a few more modifications are needed for clarity. Specific notes in regard to the submitted rebuttal answers follow a ">" symbol below.

A. In the introduction, it is stated "...we optimized the conditions to conserve PAM activity at the solubilized full-length mGlu2 receptor and measure single molecule..." It should be noted that this is a chimeric (SNAP/CLIP tagged) receptor and should be stated as such.

>> Our receptor is labeled with a SNAP-tag (not CLIP) at its N-terminus (but subunits in the receptor dimer). The sequence is now given in Fig. S25. The introductory sentence has been changes to: "Here, we optimized the conditions to conserve PAM activity of the solubilized full-length human mGlu2 receptor, N-terminally labeled through a SNAP-tag in each of the subunits, and measured single molecule Förster resonance energy transfer (smFRET) at nanoseconds time resolution."

>Acceptable for publication

Thank you, no further changes were made.

B. It is noted within the manuscript that fluorescent contaminants were present within GDN and CHS preparations. Please state the absorbance and emission maxima for these preparations. It should be stated whether these fall within or outside the measurement settings of the smFRET analysis. If overlapping, can this fluorescence contribute within the kinetic window of the assay?

>> At the concentration we used this contamination is minimal (see our answer to Reviewer 1, #4, last paragraph). Nevertheless, it might lead to the appearance of fluorescence bursts at apparent FRET efficiency close to zero and we cannot exclude that the population we observe at very low FRET efficiency partly arises from these impurities. Nevertheless, we do not draw any conclusion on the nature of this population and its presence does not change the conclusion of our study.

> No conclusions may be drawn from these states in isolation; however, at least in one analysis the very low FRET state is combined with the low FRET state which is then used to draw conclusions.

"We assigned the VHF and HF populations to a conformational ensemble representing the resting/inactive VFT state and the VLF and LF populations to the active state (Fig. 3b-h). Then, to gain a quantitative view of mGlu2 receptor activation, we calculated the fraction of active molecules found in the LF+VLF states relative to all molecules." Please see a continuation of this response in part C.

To account for this remark, we have now reanalyzed all data, comparing the fraction of active state calculated as $F = LF / (HF + LF)$ with the fraction calculated as $F_{all} = (VLF + LF) / (VLF + LF + HF + VHF)$. See our response in part C.

C. Perhaps the most interesting/confusing aspect of this manuscript is the invariable very high and very low FRET populations. As the authors stated, these populations could interconvert on a shortened time scale with those of the inactive (high FRET) and active (low FRET) pool of receptors. It seems possible that this could also represent a proportion of receptors which are not responsive in this assay, perhaps due to solubilization methods.

>> The presence of these populations is indeed puzzling and many hypotheses could be brought forward to explain their presence. We note however that in our 2014 study on the isolated mGlu2 VFTs (performed without detergent), ligand-invariant HF and LF states were observed as well. One explanation could be that they arise from alternative configurations of the SNAP-tag relative to the receptor but in our 2014 study, an experiment where the SNAP tag was replaced with the smaller ACP tag led to similar FRET distributions (Olofsson, 2014, Supp. Material).

>> We think the best evidence that these populations do not arise from solubilization artifacts comes from a very recent single molecule FRET study on SNAP-mGlu2 in live cells (using a construct highly similar to ours in Asher et al., 2021). Indeed, this study states that “Rare transitions were also observed to a FRET state of ~0.84 (Fig. 2h and Extended Data) consistent with previous reports of a minor population of high-FRET state conformations in isolated mGluR2 LBD dimers (Olofsson, 2014).) As we do not find notable changes in these VHF and VLF population in our study, we assume that transitions to/from these states are slower than the time resolution of our measurements (<5ms).

>These data from Asher et al, which track the dynamics of receptor complexes across time, suggest that the high FRET state occurs in functional receptors. These data also show though that this high FRET state is witnessed with a greater propensity with 15 μ M glutamate conditions than apo conditions (extended data figure 5C), suggesting that high FRET could be influenced by the presence of agonist and that this FRET state may not be related to an inactivated receptor state allowing it to be combined with the high FRET state thought to represent the relaxed receptor. Not enough is determined in the present study to allow for these states to be combined for analysis.

To account for this remark, we have now reanalyzed all data, comparing the fraction of active state calculated as $F = LF / (HF + LF)$ with the fraction calculated as $F = (VLF + LF) / (VLF + LF + HF + VHF)$. The results we obtained by calculating the F fraction are qualitatively identical to those obtained with F_{all} . Statistical significance of the differential effect of ligand is even higher in some cases. Data from Figures 3i, 3j, and 4f (calculated with F_{all}) are now presented as new version calculated with F. The text has been modified accordingly:

We fitted all distributions with four Gaussians and recovered similar values of E_{PR} and full width half maximum (FWHM), pointing to the fact that similar FRET states are populated for all conditions tested (Fig. S18a-b). The major changes in response to ligands were found to result from depopulation of the HF state accompanied by an increase of the LF state. Therefore, to gain a quantitative view of mGlu2 receptor activation, we calculated the fraction of active molecules, defined as the fractional amplitude of the molecules found in the LF state relative to the HF+LF states. As no notable changes were observed in the two minor populations at very low FRET (VLF, $E_{PR} \sim 0.1$, purple) and very high FRET (VHF, $E_{PR} \sim 0.87$, red) (Fig. S18c-d), these were not included in the analysis. We nevertheless verified that calculating the fraction of the active molecules as VLF + LF relative to all molecules led to similar results.

Additionally, more discussion is needed into how these potentially interconvertible states could relate to mGlu receptor dynamics. Furthermore, in the analyses from Asher et al., the high fret state appears to be observed on millisecond to submillisecond time windows. Does the percentage of receptors within this high

FRET state within the current analysis agree with what could be observed stochastically from many single molecules occupying this state so transiently? Some discussion to this point should be added.

There is no quantification in Asher et al about the duration of these states. However, since the time resolution in their experiments is 30ms (and not milli to submillisecons), they are populated for a significant fraction of time to be detected. As a matter of fact, FRET histogram in Asher et al. show a significant fraction of these states being detected, as shown in their Extended data Fig. 4f.

[REDACTED]	Extended data Fig 4f from Asher et al., showing a significant fraction of receptors populating the high FRET state (when D-A anticorrelation is taken into account in their analysis)
-------------------	--

a. Can the stoichiometry factor distinguish between single, correctly labeled Donor-Acceptor dimeric pairs and oligomers, such as a dimer of dimers, where FRET between subunits would become more complicated? If SNAP and CLIP receptors are used, which would more absolutely select for strictly labeled dimers, are the results the same as what might be found in a situation in which dimers of dimers might occur?

>> As requested by Reviewer 1, we now present the 2D histograms for all conditions in Fig. S14. Dimers of dimers with a ratio of Donor and Acceptor fluorophores unequal to 1 would have an S value intermediate between Donor-only ($S = 1$), Acceptor-only ($S = 0$) and D-A complexes ($S = 0.5$), which would therefore be centered around 0.25 for D-A-A, 0.5 for D-D-A-A, 0.75 for D-D-A). As a result of our stochastic labeling approach using two SNAP-tags (one on each protomer of the mGlu2 dimer) we should statistically detect them if they were present at elevated levels. We note that no significant populations of such species are seen around 0.25 and 0.75 and in any case such populations would be largely eliminated from our analysis, where we select only molecules with $0.30.35 < S < 0.6-0.65$.

>> Finally, D-D-A-A complexes at $S = 0.5$ would have a brightness twice the one of D-A complexes. We therefore selected this population around $S = 0.5$ and measured the average count rate per molecule of the acceptor upon acceptor excitation (insensitive to FRET) as a function of FRET efficiency. If any of these populations was composed of D-D-A-A complexes, this count rate should be doubled. We did not observe any significant change in count rate. As an example, here are the values obtained for a representative data set obtained in the presence of glutamate.

Population E range Acceptor – Acceptor count rate (kHz)

VLF 0-0.2 9.5

LF 0.2-0.4 10.2

HF 0.4-0.7 10.3

VHF 0.8-1 10.2

>> Finally, we would like to stress that at the concentration we used (~10pM receptor dimer), the presence of dimers of dimers is very unlikely.

>Do the count rates shown represent those determined from pooled averages of molecules for each determined FRET efficiency range? If so, could there be dimer of dimers still in the analysis? Is the filtering method for fluorescent molecule detection able to detect this increase in acceptor count and exclude D-D-A-A complexes? While it can be appreciated that, at such a low concentrations a dimer of dimers would be unlikely to form, they could be formed within membranes and would likely be solubilized as a D-D-A-A complex. mGlu receptors are known to maintain dimerization in non-denaturing conditions and it is not expected that dilution of receptor complexes would change this.

First, we would like to stress that while mGlu receptors are known to maintain dimerization in non-denaturing conditions, this is due to the presence of a disulfide bridge between the VFT of the dimer, and that such covalent bond does not exist between dimers of dimers, that are therefore much more prone to dissociation at picomolar concentrations.

Second, to still account for the possibility that dimers of dimers are present in the analysis, we performed the following controls :

1/ We estimated the labeling efficiency in our experiments. We plotted for a representative dataset the histogram of all molecules along the S axis, fitted with 3 gaussian.

The fraction of A-only ($S=0$), D-only ($S=1$) and Complex molecules were 0.48, 0.19, and 0.33 respectively, which correspond roughly to 60% labeling for acceptor and 40% for donor, indicating that overall labeling of the SNAP tag was 100%.

2/ Given these numbers, we can calculate what would be the shape of the S histogram if all molecules were engaged into dimers of dimers (that would lead to DDDD, DDDA, DDAA, DAAA and AAAA complexes)

It appears from this histogram that formation of dimers of dimers would have led to a significant fraction of species at $S=0.25$ and 0.75 , that were not present in our data.

3 / There are nevertheless some molecules with $S \sim 0.25$ in our data histogram, that could indeed correspond to a minor fraction of AA-DA complexes. We therefore verified the count rates for acceptor excitation / acceptor emission for the species at $S=0$, $S=0.5$, and $S=0.25$

Given our calculated labeling efficiencies, the brightness of the species at $S=0$ should be twice the one of molecules at $S=0.5$ (since molecules at $S=0$ are AA dimers). This is indeed the case, as seen for the following histograms (S vs acceptor count rate for $-0.1 < S < 0.1$ (left, corresponding to AA), vs $0.35 < S < 0.65$ (right, corresponding to DA)).

The molecules at $0.2 < S < 0.35$, if they were corresponding to dimers of dimers (DA-AA) should have a higher acceptor count rate than those at $s=0.5$ (D-A complexes). It can be seen from the following histogram that this is not the case. Therefore, these molecules do not correspond to dimers of dimers, but likely to pure dimers (see below), that appear at a lower S value due to noise or to donor photobleaching during their diffusion.

4/ As an additional control, we compared the FRET efficiencies obtained for the D-A molecules ($0.35 < S < 0.65$) and for molecules at intermediate S values ($0.2 < S < 0.35$). If these molecules correspond to dimers of dimers (AA-DA) they would have a very different FRET profile than the D-A molecules, as the expected distances between the fluorophore are likely different. Moreover, if the populations at VHF and VLF corresponded to dimers of dimers, they would be overrepresented in this ($0.2 < S < 0.35$) population. The FRET efficiencies histogram appears very similar (taking into account the noise due to the low number of molecules at ($0.2 < S < 0.35$)), therefore strongly suggesting that the population at ($0.2 < S < 0.35$) correspond to D-A species as well.

b. Other smFRET studies have described 2-3 states: high FRET (inactive, open-open), low FRET (active, closed-closed), and a “middle FRET” state (inactive, open-closed) that is transient between the active and inactive state. How do the 4 states which are proposed relate to the open-open, open-closed, and closed-closed receptor states?

>> Other studies have indeed identified a “middle FRET” state, corresponding to an “open-closed” configuration of the receptor’s VFTs. Specifically, in (Vafabakhsh, 2015) this state was populated at intermediate, subsaturating concentrations or was seen on single molecule time traces during the transition between the HF and LF state. In our case, this state was not specifically populated, as our experiments were performed at saturating concentration of ligands. Moreover, as with our technique we do not have access to smFRET time traces, we cannot specifically focus our attention on the transition step between the HF and LF state. In addition, the N-terminal FRET sensor used in previous as well as our study does not allow for a definite identification of the closed or open states of the VFTs but rather reports on the VFT intersubunit rearrangement. It will thus require additional FRET sensors to monitor the relationship of VFT closure and VFT intersubunit rearrangement.

>> Therefore, we think that the LF and HF states observed in our study correspond to the active (closed-closed) and resting (open-open) states respectively, and this is consistent with the ensemble studies performed with mutated mGlu2 receptors stabilized in either the R or the A VFT dimer states (Doumazane et al., PNAS 2013, Fig 4). The additional VHF and VLF states we observed in our study, where all molecules are considered and displayed, might have been present in other studies, but could have been overlooked as they represent minor populations that are not influenced by

ligand and probably not interconverting with the other states at fast timescales. Moreover, the VLF state can wrongly be assigned to a donor-only state, as alternating laser excitation was not used in these other studies.

>>Further investigation, using for example alternative labeling strategies using smaller tags such as

UAAs and other labeling positions will be required to identify the nature of these additional states.

>It is of the opinion of this reviewer that the relation of observed FRET states to Open-Open and Closed-Closed states should be stated clearly in the discussion in order to relate these findings more clearly to other mGlu literature.

We have now included a reference to this states in the discussion, as follows :

Our data reveal the presence of four VFT states. Two of them - the HF/inactive and LF/active states - are predominantly populated in a ligand-dependent manner (Fig. 3, 5 and S16), Based on previous studies^{25,30}, we attributed the LF state and the HF states to conformations in the “active” and “resting” orientations of the dimeric ECD respectively, with both VFTs in the “closed” and “open” conformations (Acc and Roo), respectively. Two minor populations (VHF and VLF) were barely affected by ligands (Fig. S18 a and b, respectively), but we note that they could be in exchange with the two major populations at timescales slower than the resolution of our method (> 5 ms). Interestingly, in a very recent study using a construct highly similar to ours, the structural dynamics of mGlu2 were monitored using smFRET at the surface of living cells, with a lower time resolution (40ms)⁴⁶. Transition from a high FRET ($E = 0.44$) to a low FRET ($E = 0.3$) state were observed upon glutamate binding. But notably, some transitions to a very high FRET state ($E = 0.88$) and to a lower FRET state (not reported by the authors) could be observed in some traces. We hypothesize that this seldom populated VHF and VLF states could correspond to the ones observed in our study, although the determination of their exact structural nature would require further studies.

c. Within the literature, smFRET studies of mGlu receptors have been performed using integration times of 30-100 ms. This time is longer than the 5 ms time scale of the assay employed in this study. The very high fret and very low peaks are not present within these studies. The argument here is presumably that the increased specificity in integration time has captured a new state which is missed by other methods, much like the interconversion mentioned in the 0.2ms integration in Figure 4D. Have the authors performed an experiment with an increased integration time?

>> As stated above, it is possible that the VLF and VHF were present in previous studies, but have been overlooked. One argument for this hypothesis is that VLF and VHF states have been transiently observed using single molecule FRET in living cells by the Blanchard/Javich group recently (Asher 2021).

>>In our case, it is unfortunately not possible to observe mGlu conformational dynamics for longer time periods. Our observation time is limited by the diffusion time of each single receptor through the confocal volume. An experiment with longer integration time would require mGlu2 immobilization without disturbing its structural dynamics together with confocal scanning for high temporal resolution measurement. This type of experiment would require a lot of optimization, control and instrumental development and goes beyond the scope of the present work.

>We agree that these additional experiments would be beyond the scope of this study, but please see response to part B, C.

We have now addressed the reviewers concerns in part B,C.

Can the data be pooled after the experiment to more accurately compare to previously described high and low FRET states? This should be clearly discussed.

>> It is not clear to us what the reviewer means by “pooling” the data. For each condition, our histograms and dynamic analysis already come from the pooling of thousands of molecules but the observation time for each molecule remains limited to around 5 ms.

>Originally, the reviewer was under the impression that the integration time could be increased to capture a distribution that more closely resembled other studies, suggesting more comparability between this and other studies. It is now apparent that this is dependent on the current experimental design, and so the authors' response is understood.

Thank you, no further changes were made.

D. In Figure 4 a-c, is there a more quantitative way to represent the “skew”, i.e., residuals?

>> The representation of Figure 4 is the one classically used in the literature to represent this type of data. Nevertheless, we agree with the reviewer that a representation that highlights the deviation of the data from the static FRET line might more efficiently underline the differences observed between the ligands. We have therefore generated such a representation, now presented in Fig. S20, which nicely supports our conclusions obtained from the visual observation of Figure 4 a-c.

>Acceptable for publication

Thank you, no further changes were made.

E. Additional clarity is needed in the experiments involving antagonists. In Figure 3H, the addition of antagonist does not alter the receptor state distribution, contrary to Vafabakhsh 2015 where the low FRET state was abolished by LY34. These previously published data make sense if closure of the LBD is needed for reorientation of the LBDs relative to one another. This also agrees with data from Doumazane et al 2013 in which agonist was required for transition to the activated, low FRET LBD state in full length receptors, and which was blocked entirely by addition of LY34. The data within the current manuscript suggest that LBD reorientation may still take place while the LBD open confirmation is stabilized by antagonist. This would suggest that agonist binding and LBD transition are effectively separate processes. It could be argued that these data, along with the presence of very high and very low FRET states, may represent an artifact specific to the experimental system and should be specifically addressed within the discussion.

>> The reviewer is right in pointing out that we do not see any effect of the antagonist LY34 on the Apo receptor. But we would like to point out that in the study of Vafabakhsh 2015, such an effect is not seen either in the case of mGlu2. It is only observed in Vafabakhsh 2015 when mGlu3 is monitored, a subtype that is reported to have a higher basal activity than mGlu2, and a much higher glutamate affinity (Tora et al. 2018), making it more sensitive to tens of nanomolar

glutamate contamination in the media. In the case of Doumazane 2013, with LRET experiments performed on mGlu2 in cells, the observed antagonist activity probably comes from the presence of residual glutamate in the buffer that is displaced by LY34.

>> We note that in our experiments on membrane fractions, where some residual glutamate might be present, a slight effect of LY34 is observed (Fig. S12). Nevertheless, to demonstrate the effect of LY34 in our single molecule conditions, we have now introduced a new supplementary Figure, where we show that LY34 at 1mM displaces glutamate used at 100 μ M and therefore reverses its effect (Fig. S17).

>> Accordingly, we have modified the main text as follows.

“Finally, we noted that application of saturating concentrations of the competitive orthosteric antagonist LY341495 (LY34) led to a similar distribution as seen for the apo receptor (Fig. 3h). This antagonist is able to bind and properly reverse the effect of a subsaturating concentration of glutamate (100 μ M) (Fig. S17). This indicates that no basal receptor activity or residual Glu was observed in our preparations, which was further verified by titration with LY34 in LRET measurements (Fig. S13a-e).”

>We understand the author’s argument about antagonist blocking the effect of subsaturating glutamate. We agree that this assay seems to have low basal activity. However, even without glutamate present, antagonist will bind to the LBD and stabilize the open conformation. One would expect, if LBD closure is inherently linked to subunit rearrangement and alteration in FRET efficiency that this would result in an alteration in the distribution of FRET states, specifically with a reduction in the very low and low FRET states. The absence of this effect suggests a rearrangement that is independent of LBD closure and should be discussed.

The SNAP-based FRET sensor we are using reports mainly on the rearrangement of the dimer from the “resting” to the “active” state. This was established in the seminal study of Doumazane et al. (ref. 30). In the Figure 1 of this paper, it was shown using the available crystal structure of the ECD available that the distance between the N-termini depends on the reorientation of the dimer, and not on the opening or closure of the VFTs. Therefore, the antagonist-bound VFT dimer, with both VFTs maintained in an open conformation, may possibly exist in both a resting and an active orientation (see crystal structure of the mGlu1 VFT dimer with bound antagonist, observed in the Aoo state). However, our data show that most single molecules are in a “high FRET” state, corresponding to the resting orientation of the VFT dimer.

Accordingly, we have now modified the sentence describing the effect of the antagonist as follows.

“Finally, we noted that application of saturating concentrations of the competitive orthosteric antagonist LY341495 (LY34) led to a similar distribution as seen for the apo receptor (Fig. 3h), consistent with a receptor remaining in the resting state³⁰.”

F. How do these data relate to PAMs which bind the TMD but display agonism as well as allosteric activity? It is important for this technique to establish that differences observed in full length receptors by more traditional methods with unique allosteric modulators can be corroborated (or refuted) in this methodologic paradigm.

>> It would indeed be interesting to explore the structural dynamics of mGlu in response to other molecules that bind to the 7TM domain but exert by themselves an agonist effect in the absence of orthosteric agonist (molecules so called Ago-PAM), Unfortunately, although these types of Ago-PAM are available for other mGlu subtypes such as mGlu4 (Rovira et al., FASEB J 2015) or mGlu5 (Noetzel et al., Mol Pharmacol 2012), such type of compounds have not been described and validated to our knowledge in the case of mGlu2. Such an investigation on other mGlu subtypes will be the topic of further studies.

>Acceptable for publication

Thank you, no further changes were made.

G. The Supplemental section is extremely confusing. For example, the Ro64 compound is mentioned in Supp Figure 9, but not discussed in the text in reference to this figure. Similarly, DCG-IV and Ro64 are also used in Supp Figure 12, but data with these compounds is also not discussed in the text. A discussion regarding Supplemental Figure 15 appears before Supplemental Figure 13; similarly, Figures S17-18 are discussed before Figure S16. While the Gi condition in Figures S17 and 18 is mentioned, the other conditions tested in Figures S17-18 are not discussed. There is also a Figure S19 discussed in the test that is not in the Supplemental Figure file. It is also suggested to move Figure 5c to the end of Figure 5 due to the flow of the results section regarding this figure. The order of many figures should be rearranged, and it is suggested that the authors should walk through each panel, in order, with each experimental condition discussed in the main body of the paper. With 19 supplemental figures, a clear presentation becomes even more imperative. The authors (end

editors) should also consider whether the figures pertaining to the distinct detergent conditions could be consolidated into a single table, perhaps with two figures that show divergent effects of detergents left in as examples.

>> We thank the reviewer for pointing out these discrepancies and errors and apologize for the confusion it might have generated. As we have added new Supplementary Figures, this section and the corresponding references in the text, have been completely renumbered, and we have carefully checked that the order and the references to the Figures are consistent. Concerning the presentation of the data for the detergents as curves instead of as a table, we would like to insist on providing the raw titration curves, which show the quality and the reproducibility of our data.

>> We have added descriptions as follows:

“Under these detergent conditions, a small but significant effect of the negative allosteric modulator (NAM) Ro64-5229 that reduces the pEC50 of glutamate in membranes (Fig. S12a-c) and live cells 30, was observed as well (Fig. S9a-c). In addition, we confirmed the effect of the partial agonist DCG-IV, previously shown to promote changes in the VFT intersubunit orientation to a lower extent than full agonists 30. Such partial effect, strongly potentiated by the addition of BINA, was indeed observed on the solubilized receptor (Fig. S13). Notably, the effects of DCG-IV in the absence or presence of NAM and PAM reflected those observed in crude membranes (Fig. S12d-f), thus confirming native-like ligand responsiveness of the detergent-solubilized receptor.”

“This effect was reversible, as addition of an excess of the NAM Ro64 to receptors after activation by

500 nM BINA + Glu decreased the fraction of active receptor to a similar level observed when only Glu + NAM were applied (Fig. 5i-j and S16b), being slightly below that observed in the presence of Glu alone (Fig. 3i).”

“Application of the NAM Ro64 in the presence of Glu did not alter these partially remaining dynamic behavior (Fig. S21c and 22c).”

>> We have further adapted various figures with regard to coherence and readability by increasing font size, displaying the same type of cartoons and logical order of the panels. Further we have made many corrections to the main text and figure legends with regard to grammatical errors, coherence for abbreviations used and renumbering of figures. All figures displayed should now be mentioned in the text. All changes made are highlighted in yellow.

>Thank you for making these corrections.

Thank you, no further changes were made.

In the Figures, there are no statistical comparisons between any of the conditions. These need to be included; again, a table may be a better way to represent all of the data in the supplemental figures showing the various detergent conditions.

>> We think that there are a few Figures where a statistical analysis might be required to underline the effect (or the lack of effect) of different ligands or experimental conditions, namely Figure 5i, S9, S12, and S13. Therefore, for some conditions reported on these Figures, where only a small difference between data was observed, we performed and now report a one-way ANOVA with Sidak or Turkey's multiple comparisons test as indicated. The results of these tests are consistent with our interpretation of the data as reported in the main text.

>There are still figures where statistics are not clearly marked. Statistical tests for each figure should be clearly denoted within the figure legend.

Ex. In figure 2 there is no mention of any statistical tests to denote differences in membrane preparations.

Ex. S23. It is noted that 95% confidence intervals are plotted but there is no information on what statistics were run or if any of the observed differences are significant.

We have now included appropriate statistical test for the data shown in Figures 2a-f and Supplementary Figures 1-13, 23 and 24 as indicated in the corresponding Figure legends. Additionally, we have recalculated the errors for all pEC50 plots, which are now given as standard deviation.

Reviewer #3 (Remarks to the Author):

The authors have addressed my concerns.

Thank you, no further changes were made.

REVIEWERS' COMMENTS

Reviewer #1 (Remarks to the Author):

It is unfortunate that the authors are reliant on software that doesn't allow them to plot and analyze the 0.2 ms binning data in terms of the actual data but only normalized. It is helpful to show the numbers in the legend but it is difficult to assess whether these data are truly robust - other than the authors claim that 900 events means it is robust. It is more than an order of magnitude less populated than the 1 ms binning and there still is a chance that the inferences are incorrect and based on an artifact without sufficient data. Some caveat seems called for here about the 0.2 ms data being much less populated and the value of future study and verification. The inferences will nonetheless help guide other research and time will tell on this additional state.

Reviewer #2 (Remarks to the Author):

We thank you for the opportunity to review Nature Communications manuscript NCOMMS-21-05720C.

At this stage of the review process, the authors have sufficiently and extensively addressed the initial concerns of the original manuscript. I thank the authors for their substantial, detailed revisions.

I recommend this article be accepted for publication.